# M3GIA: A Cognition Inspired Multilingual and Multimodal General Intelligence Ability Benchmark

## Abstract

As recent multi-modal large language models (MLLMs) have shown formidable proficiency on various complex tasks, there has been increasing attention on debating whether these models could eventually mirror human intelligence. However, existing benchmarks mainly focus on evaluating solely on task performance, such as the accuracy of identifying the attribute of an object. Combining well-developed cognitive science to understand the intelligence of MLLMs beyond superficial achievements remains largely unexplored. To this end, we introduce the first cognitive-driven multi-lingual and multi-modal benchmark to evaluate the general intelligence ability of MLLMs, dubbed M3GIA. Specifically, we identify five key cognitive factors based on the well-recognized Cattell-Horn-Carroll (CHC) model of intelligence and propose a novel evaluation metric. In addition, since most MLLMs are trained to perform in different languages, we go beyond English to encompass other languages, including Chinese, French, Spanish, Portuguese and Korean, to construct our M3GIA. We make sure all the data relevant to the cultural backgrounds are collected from their native context to avoid English-centric bias. We collected a significant corpus of data from human participants, revealing that the most advanced MLLM barely reaches the lower boundary of human performance in English, and there remains a pronounced disparity in the other five languages. Importantly, we found that designing IQ tests for MLLMs is crucial, as the evaluation of M3GIA achieves a significantly stronger alignment with human preferences compared to traditional task-oriented benchmarks. Moreover, grounded in CHC theory, we discovered that the number of samples seen by the vision encoder has a greater influence on the model's visual capabilities than its parameter size.

## 1 Introduction

In 1956, researchers across different domains, including mathematics, cognitive psychology and computer science, pointed out an interesting direction, dubbed artificial intelligence (AI). The formal definition is *"The study is to proceed on the basis of the conjecture that every aspect of learning or any other feature of intelligence can in principle be so precisely described that a machine can be made to simulate it." (McCarthy et al., 2006).* Through extensive efforts in pursuing artificial intelligence, the field has converged to a paradigm of data-driven machine learning models, which are still deeply intertwined with cognitive science as they often mirror basic cognitive mechanisms, e.g. convolutional neural networks (Krizhevsky et al., 2012) and the attention mechanism (Vaswani et al., 2017). Recent advances, such as GPT-4o (OpenAI, 2024), demonstrate that these MLLMs can outperform human on various complex tasks (Achiam et al., 2023; Wang et al., 2023b) and shed light to emergent ability with the increasing scale of data and model size (Wei et al., 2022). In light of these developments, our aim is to evaluate these state-of-the-art models through the lens of cognitive science, as it directly aligns with the primary motivation of AI research.

To explore the mental intelligence emerging from these large models, efforts have been directed toward analyzing these models from a psychological perspective. Some pioneering works report that LLMs have demonstrated human-like cognition (Binz & Schulz, 2023; Kosinski, 2023b). For instance, Theory of mind (ToM) has been applied to assess large models, revealing that GPT-4 exhibits ToM capabilities similar to human inference patterns (Bubeck et al., 2023; Kosinski, 2023b).

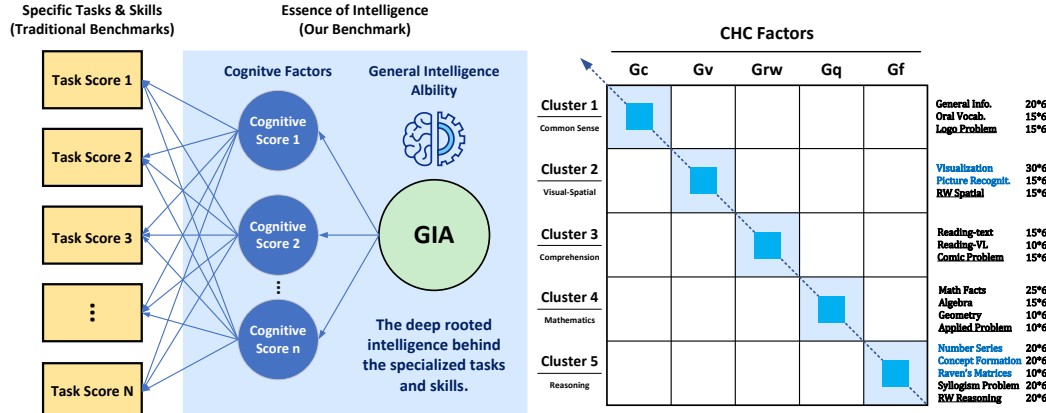

Figure 1: **Overview of multi-lingual multi-modal general intelligence ability benchmark. (Left)** In contrast to traditional benchmarks that focus on evaluating specific task performances, we draw inspiration from cognitive science to categorize five cognitive factors, try to provide a feasible evaluation of general intelligence ability (GIA). **(Right)** Specifically, we adopt the factors from the CHC theory to disentangle fundamental cognitive abilities with existing evaluation tasks. In addition, to further understand how language impacts such ability, we collect or design questions in six languages with large population.

Meanwhile, Multimodal Large Language Models (MLLMs), which use powerful LLMs as brain to process and integrate multimodal information, have exhibited impressive emergent abilities, such as generating website code from images (Zhu et al., 2023), understanding the meaning of a meme (Yang et al., 2023), and math reasoning (Driess et al., 2023). Thanks to their ability to process information from a broader spectrum of sources, they exhibit a more holistic cognitive process, resembling human cognition more closely than models confined to purely linguistic input.

Existing multi-modality benchmarks, such as MMBench (Liu et al., 2023b), MME (Fu et al., 2024), and MM-Vet (Yu et al., 2023), have made the attempt to compartmentalize model capabilities across multiple tasks. For instance, MMBench covers 20 different abilities, encompassing function reasoning, physical property reasoning, object localization and social reasoning. However, they often fail to provide a persuasive explanation for their selection of dimensions, as they tend to be mired in subjectivity and lack a solid theoretical underpinning. Moreover, as depicted in Figure 1 (left), their ability dimensions are still rather task-oriented, neglecting a systematic evaluation of the models' underlying cognitive abilities that govern task performance through the lens of cognitive science. This oversight raises concerns that benchmarks might devolve into mere training targets rather than instruments for true insight, failing to provide a holistic measure of the models' capabilities (Schaeffer et al., 2024). In short, the ability to solve specific tasks is insufficient to reflect the true level of intelligence, as supported by a psychological study(Poldrack & Yarkoni, 2016), and formally evaluating the cognitive factors of MLLMs remains largely unexplored.

In this paper, we close the gap by introducing the first benchmark that comprehensively evaluate the cognitive abilities of MLLMs under the theoretical umbrella of the well-recognized Cattell-Horn-Carroll (CHC) Model of Intelligence (Schneider & McGrew, 2012), dubbed M3GIA. As in Figure 1(right), based on the CHC Model, we categorizes the cognitive capacities of current MLLMs into five dimensions: Fluid reasoning (Gf), Comprehension-Knowledge (Gc), Visual processing (Gv), Reading and Writing (Grw), Quantitative knowledge (Gq), and collect corresponding questions as a measurement. In addition, as using multi-lingual data to scale up the capability of MLLMs becomes a de-facto standard, we are curious whether languages make any impact on their cognitive abilities. As such, we extend our benchmark to include five more languages, including Chinese, Spanish, French, Portuguese and Korean roughly based on their population, to disentangle the language factor with cognitive ability.

To evenly assess the five cognitive dimensions, we refer to human intelligence tests, such as the Raven's Progressive Matrices Test (Raven, 2003) and the Woodcock-Johnson IV Tests of Cognitive Abilities (WJ IV) (Schrank & Wendling, 2018), and establish broad question types that correspond to the these cognitive dimensions, which are further subdivided into 18 narrow question types (see later Sec. 3). Overall, our M3GIA includes 1.8K high-quality meticulously human-annotated questions,

with over 73% created by professionals due to the non-public nature of human intelligence tests. This prevents potential data leakage from directly collecting extensive data from existing sources. On the other hand, it makes the construction of M3GIA labor-intensive and expensive. The test for each language maintain consistency in terms of the number of questions, structure, and distribution of question types. In addition, to highlight the multilingual nature of our benchmark, we collect data relevant to cultural backgrounds from native language sources rather than simply translating them from English, thereby avoiding the English-centric bias.

We evaluate 24 MLLMs, including the state-of-the-art close and open-sourced ones. In general, The latest advancements in MLLMs have achieved performance levels that fall within the lower boundary of human intelligence in English. Yet, there remains a pronounced disparity in the other five languages assessed. We also notice that MLLMs' proficiency in one cognitive domain often translates into superior performance across other domains as well. This phenomenon interestingly aligns with the pattern observed in human intelligence which empirically suggests the existence of General Intelligence Ability (GIA) in MLLMs.

## 2 RELATED WORKS

**Evaluation Benchmark for MLLMs.** As multimodal large language models (MLLMs) exhibit remarkable generalization capabilities across a broad spectrum of downstream tasks, relying exclusively on their performance within single vision-language tasks — such as visual recognition (Goyal et al., 2017), image description (Chen et al., 2015; Agrawal et al., 2019; Young et al., 2014), scene text understanding (Singh et al., 2019; Sidorov et al., 2020), and external knowledge (Marino et al., 2019) — is insufficient to fully uncover the comprehensive performance of MLLMs. People then turn to a new paradigm to construct all-round benchmarks to assess a broader spectrum of challenging multimodal tasks (Yin et al., 2024; Xu et al., 2023; Li et al., 2023; Liu et al., 2023b; Fu et al., 2024; Yu et al., 2023). Another trend in MLLM assessment is the use of human exam questions (Lu et al., 2023; 2022; Zhong et al., 2023; Yue et al., 2023; Zhang et al., 2024). For instance, AGIEval (Zhong et al., 2023) sources questions from standardized exams such as college entrance exams and lawyer qualification tests. While these benchmarks makes progresses in evaluating the human-centric ability of MLLMs, it may not be suitable to evaluate the intelligence of MLLMs because research in psychological field points out that the superficial performance on tasks alone cannot be a solid indicator for human's intelligence.(Poldrack & Yarkoni, 2016)

**General Intelligence Ability and the CHC Theory.** Arising from the empirical fact that an individual's proficiency in one area frequently correlates with high performance in other areas, Charles Spearman first introduced General Intelligence Ability (GIA) in 1904 (Spearman, 1961). This construct refers to the idea that a single underlying factor, known as the g-factor, can account for the positive correlations among cognitive abilities and reflect the general intelligence that fundamentally underlies an individual's intelligence. To concretely understand GIA, numerous attempts has been made to model the structure of human cognition. John Carroll's Three-Stratum Model (Carroll, 1993) elaborated on this with a hierarchical structure of intelligence, including a general "g" factor and specific cognitive abilities. Howard Gardner's Multiple Intelligence Theory (Flynn, 1987) proposed diverse forms of intelligence, while Sternberg's Triarchic Theory (Sternberg, 1985) focused on practical, creative, and analytical aspects. These theories collectively contributed to the development of the Cattell-Horn-Carroll (CHC) model of intelligence, which is the most comprehensive and empirically validated structural model of human cognition (McGrew & Evans, 2004) to date, integrating various aspects of cognition into a unified framework.

**Comparison with Existing Psychology-inspired Benchmarks.** Significant strides have been made to explore LLMs' capabilities using psychological tools. These efforts, however, predominantly concentrate on aspects such as social reasoning, emotional abilities, values, and personality. In contrast, M3GIA's main contribution lies in its commitment to providing the first "IQ test" for MLLMs, focusing on pure intelligence rather than other psychological dimensions.

- **ToM benchmarks** (Jin et al., 2024; Kosinski, 2023a; He et al., 2023; Gandhi et al., 2024)**:** Theory of Mind (ToM) is the ability to understand other's mental states based on their observed behavior (what someone else is thinking or feeling). It is a hallmark of "social intelligence" that fall within the scope of Social Quotient (SQ), which is a distinct realm from Intelligence Quotient (IQ)

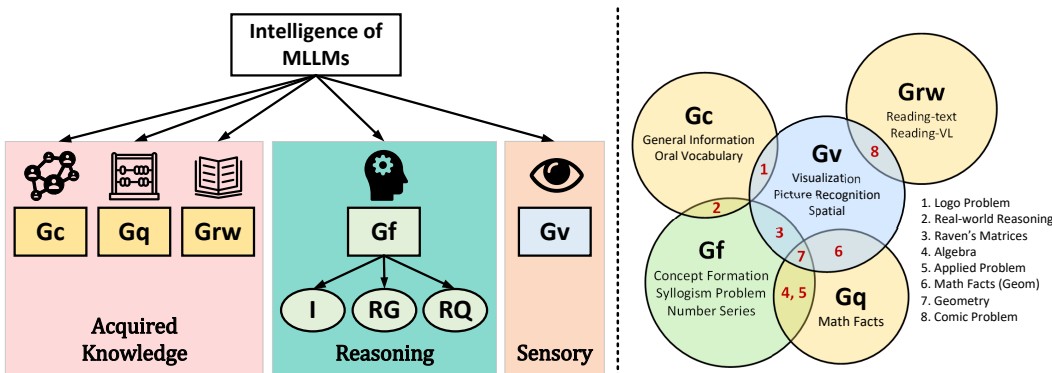

Figure 2: **Structure of our CHC inspired model of cognitive abilities. (Left)** We identified five key cognitive factors for current MLLMs: Comprehension-Knowledge (Gc), Quantitative knowledge (Gq), Reading and Writing (Grw), Fluid reasoning (Gf), and Visual-spatial processing (Gv). In the hierarchical structure, Gf is further subdivided into three narrow factors: I (Induction), RG (Deductive Reasoning), and RQ (Quantitative Reasoning). **(Right)** A conceptual map of the five cognitive factors and their overlaps with each other.

and Emotional Quotient (EQ) in psychology studies. ToM's independence to intelligence is also validated in (Rajkumar et al., 2008).

- **Other psychology-inspired benchmarks:** PsychoBench (Huang et al., 2023) divides psychological measurement into PERSONALITY TESTS and ABILITY TESTS, with ABILITY TESTS subdivided into Knowledge&Skills, Cognitive, and Emotional. However, its ABILITY TESTS only includes the Emotional Abilities Test and doesn't include Cognitive Abilities test (target of M3GIA). As the paper states: *"Intelligence Quotient (IQ) tests ... represent one of the most comprehensive, intricate, and renowned evaluation tools in this category (cognitive tests). However, since these assessments often incorporate visual elements unsuitable for LLM evaluation, this aspect remains a potential avenue for future investigation."* Psychometrics Benchmark (Li et al., 2024b) advocates for a comprehensive psychological measurement for LLMs, including personality, values, emotion, ToM, motivation, and intelligence. However, they didn't complete the 'intelligence' part and only discussed its potential. CogBench (Coda-Forno et al., 2024) is a task-oriented benchmark that focuses on decision-making tasks, such as long-term rewards. Its tasks are highly composite, often requiring not only intelligence but also value-based judgments. For example, temporal discounting indicates whether an agent prefers smaller but immediate gains over larger delayed ones, while BART is used to assess risk-taking behavior. However, the tasks are too high-level to be used as a means of evaluating the basic factors of intelligence (as they are hard to disentangle and attribute).

## 3 M3GIA

Concretely, we introduce the first cognition inspired multi-linguistic and multi-modal benchmark to evaluate the general intelligence accuracy of large models. In short, our M3GIA distinguishes itself from existing benchmarks as follow:

- **Cognition Inspired:** In contrast to existing benchmarks that focuses on task-level evaluation, we study the intelligence of large models from a cognition perspective. The benchmark dissects the cognitive abilities of contemporary MLLMs into five foundational factors, as per the Cattell-Horn-Carroll theory. This cognitive theory underpins the structure of our evaluation, informing the specific types of questions devised to test each cognitive skill.

- **Multilingual Coverage:** To comprehensively measure the cognitive abilities of multimodal large models across multiple languages, M3GIA is constructed to span six languages: English, French, Chinese, Spanish, Portuguese, and Korean. In order to mitigate English-centric bias, all data relevant to cultural backgrounds have been sourced from native language resources, except for questions that transcend cultural considerations—such as the Raven test and number series problems.

## 3.1 THE FIVE-FACTOR COGNITIVE MODEL OF M3GIA

To formally study the intelligence level of MLLMs, we start from the state-of-the-art cognitive model, Cattell-Horn-Carroll (CHC) (Schneider & McGrew, 2012), which is by far the most empirically validated structure model of human cognition (McGrew & Evans, 2004). The CHC theory articulates a hierarchical framework of human cognitive abilities divided into three strata: general intelligence "g" (stratum III), broad cognitive abilities (stratum II), and narrow abilities (stratum I). While there is ongoing discourse regarding the exact delineation of stratum I, stratum II have achieved substantial consensus and are well-supported by empirical evidence and practical application (Caemmerer et al., 2020). These include Fluid Reasoning (Gf), Comprehension-Knowledge (Gc), Visual Processing (Gv), Auditory Processing (Ga), Short-term Memory (Gsm), Long-term Retrieval (Glr), Processing Speed (Gs), Quantitative Knowledge (Gq), and Reading and Writing Abilities (Grw). These broad but domain-specific abilities are nevertheless positively associated with one another. This positive manifold is accounted for in the CHC model by a general factor of intelligence ("g") at stratum III.

It is important to note that an intelligence test doesn't need to encompass all CHC factors to be effective. Rather, it should strategically select a relevant subset of Stratum II factors tailored to the specific target of the test. For instance, the Stanford-Binet Intelligence Scales (Roid & Pomplun, 2012) focus on five specific factors (Gc, Gf, Gq, Gv, Gwm), while the WJ IV Tests of Cognitive Abilities (Schrank et al., 2016) incorporate seven (Gc, Gf, Gv, Gwm, Gs, Glr, Ga).

As shown in Fig. 2, the structure of our M3GIA is underpinned by the five-factor hierarchical cognitive model, which is derived from the CHC model of cognitive abilities. Given that the majority of current MLLMs are not yet expanded to embrace the auditory modality, we have not included the Ga (Auditory) factor in this version of M3GIA, reserving it as one of the directions for future expansion. Consequently, based on the consultations with psychology experts, we have chosen to assess the cognitive abilities of current MLLMs in this iteration of M3GIA by focusing on five key CHC factors: Gc, Grw, Gq, Gf, and Gv. The selection of the five factors is also well-supported by psychological validation through factor analysis (Phelps et al., 2005), which shows that Gf (0.98), Gq (0.87), Gc (0.79), and Gv (0.68) have the highest significant factor loadings related to general intelligence, while Ga and Gs only have loadings of 0.47 and 0.48. We provide a more detailed discussion on why these five factors were chosen to evaluate MLLMs in *Appendix A.3*.

Interestingly, the five factors we select align closely with those of the renowned Stanford-Binet Test, Fifth Edition (SB5) (Roid & Pomplun, 2012), which was also constructed upon five cognitive factors derived from the CHC theory. Specifically, the five cognitive factors identified in the SB5 are: Fluid Reasoning (FR), Knowledge (KN), Quantitative Reasoning (QR), Visual-Spatial Processing (VS), and Working Memory (WM). Except for Working Memory (WM), which we have substituted with Grw, these factors align directly with our selected factors, corresponding to Gf, Gc, Gq, and Gv, respectively. This alignment is noteworthy, as the selection of these factors for the SB5 was based on extensive research on school achievement and expert ratings of the importance of these factors in the assessment of reasoning, especially in giftedness assessment (Roid & Barram, 2004).

## 3.2 QUESTION DESIGN AND COLLECTION

Our M3GIA contains a total of 1.8K multiple choice problems, of which 1,200 are Visual Question Answering (VQA) questions. To prevent potential data leakage and given that human intelligence tests are not publicly available, 73% of our data is manually crafted by ourselves, while the remaining 27% is sourced from existing materials, following the practices of MMMU (Yue et al., 2023) and M3Exam (Zhang et al., 2024), which also derive their data from existing human exams. The dataset size of M3GIA was carefully determined by balancing several key considerations:

- **Human Baseline:** M3GIA depends on human data to construct the GIA model, which requires reliable human baseline measurements. Research by Converse & Presser (1986) indicates that prolonged tasks can degrade response quality, underscoring the importance of balancing comprehensiveness with practicality. To minimize the number of questions while ensuring validity, we determine the number of questions based on findings from Burisch (1997), which revealed that in cognitive assessments, extending a scale beyond a certain limit can actually undermine its validity. Interestingly, the validity plateaus when the number of items in a subtest hits 15. Considering our 18 subtests, we settled on incorporating 300 questions per language (>$15 \times 18 = 270$) to

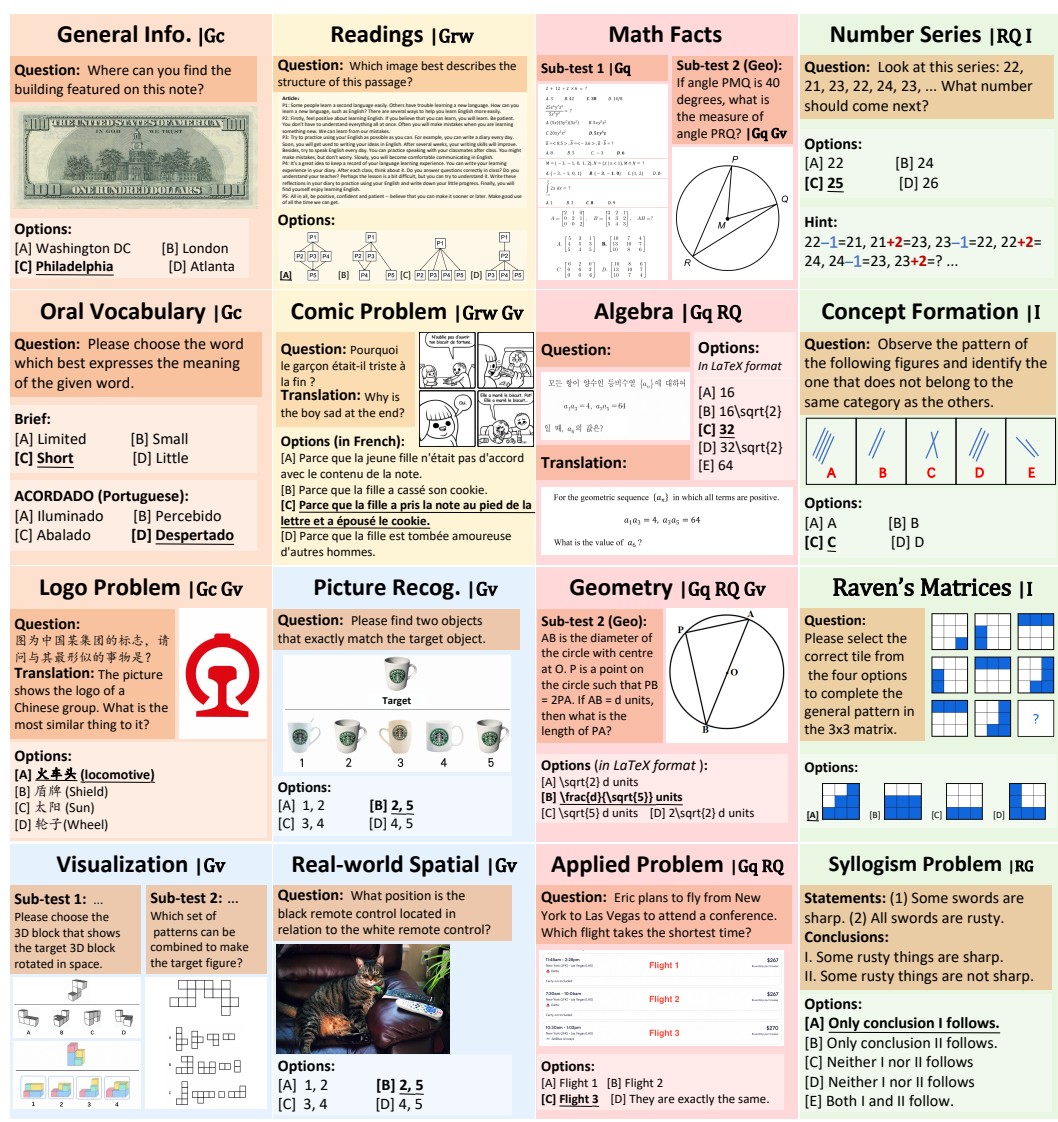

Figure 3: **Questions overview of M3GIA.** To assess the five CHC cognitive factors—Gf, Gc, Gq, Grw, Gv correspondingly—we devised five broad question clusters: common sense (orange), visual-spatial (blue), comprehension (yellow), mathematics (red), and reasoning (green). To prevent the assessment of any particular ability from being constrained to a fixed and singular perspective, we have stratified each of the five clusters into 2-4 specialized narrow question types, each narrow question type reflects a different perspective on the broad CHC ability. This subdivision results in a total of 18 subtasks. All the QAs are in the format of multiple choice problems whose answers are marked [A][B][C][D].

guarantee a thorough evaluation. Typically, participants require six hours to tackle 300 questions, often necessitating a whole day to complete the full task.

- **Benchmark Alignment:** The data volume aligns with established cognitive benchmarks. For instance, the U.S. Human Connectome Project's Spatial Orientation task includes 24 trials, while the WJ-IV test typically comprises 10-30 questions per task.

**The Selection of Question Types.** Each of our question type is specifically crafted according to the definition of the corresponding CHC factor. To ensure M3GIA maintains professionalism as a cognitive science test, the question types selected for each CHC factor closely adhere to the designs of the well-recognized WJ-IV (Schrank & Wendling, 2018). These include: Concept

Formation, Number Series, Reading (Mather & Wendling, 2015), Math Facts, General Information, Oral Vocabulary, Visualization, and Picture Recognition, while the remaining question types were created in collaboration with psychology experts. A detailed description of the question types and their corresponding CHC factor definitions is provided in Table 2 and Table 3 in *Appendix.D*. For example, a key facet of Gv is the ability to perceive complex patterns and mentally simulate transformations such as rotation, resizing, or partial occlusion (Schneider & McGrew, 2012). In our Visualization tasks, the model is asked to identify the rotated 3D block that matches the original or to identify pieces that form a complete target shape. Please refer to *Appendix D* for detailed correspondence between the CHC factors and our question types.

As shown in Fig. 1, we have devised five broad question clusters: reasoning, visual-spatial, common sense, mathematics and comprehension, separately corresponding to the assessment of the five CHC cognitive factors – Gf, Gv, Gc, Gq, and Grw. (See *Appendix* for detailed definition of the factors.) To prevent the assessment of any particular ability from being constrained to a fixed and singular perspective, we have stratified each of the five clusters into 2-4 narrow question types that reflect different perspectives on a broad CHC construct. This subdivision results in a total of 18 distinct question types, each designed to tap into different facets of the ability being measured.

Moreover, as illustrated in the right part of Fig. 2, the five cognitive factors are not isolated but rather overlap with each other. For example, Fluid reasoning (Gf) not only has a process facet (inductive vs. deductive reasoning) but also has a content facet (verbal, spatial, and quantitative), each of which overlaps with other broad abilities (Schneider & McGrew, 2018). In order to conduct a comprehensive measurement of this overlapping nature, our narrow question types include not only tests that measure each cognitive factor individually but also cover the parts where these factors overlap. The corresponding relationships between the question types, the cognitive factors and their intersections are also shown in Fig. 2.

We provide detailed explanations in the appendix, including More Details on Data Collection and Annotation (*Appendix A.2*), Introduction to the Evaluation Questions (*Appendix C*), and Data Curation Process (*Appendix E*). In Data Curation Process, we discuss key aspects such as data balancing, quality control, accuracy checks, and the management of question variance across different languages.

## 3.3 METRICS

We use two type of metrics in our evaluation benchmark. For each question type, we follow the existing benchmarks (Liu et al., 2023b; Fu et al., 2024) to use accuracy. However, to holistically compare the cognitive ability, we design a novel metric general intelligence accuracy (GIA) based on findings in cognitive field. To compute the GIA scores of the models, we adopted a standard psychometric approach. This involved utilizing a confirmatory factor analysis (CFA) model, developed from our collected human evaluation data. See *Appendix F* for more details about the CFA process.

## 4 EVALUATION RESULTS

In this section, we evaluate a total of 24 MLLMs and 480 human participants using our M3GIA. The MLLMs comprise both closed-source models, such as GPT-4o (OpenAI, 2024), and open-source models (Liu et al., 2023a; Li et al., 2024a; Young et al., 2024; Bai et al., 2023; Wang et al., 2023a; Lu et al., 2024; Chen et al., 2023), including LLaVA (Liu et al., 2023a) and Mini-Gemini (Li et al., 2024a). Our evaluation for the MLLMs is conducted under a zero-shot setting to assess the capability of models to generate accurate answers without fine-tuning or few-shot demonstrations on our benchmark. For all models, we conduct prompt engineering on the validation set and use the most effective prompt for the zero-shot setup in the experiments. All experiments are conducted with NVIDIA A800 GPUs (Liu et al., 2023a; Li et al., 2024a).

**Human Performance Baseline.** To establish a reference for human cognitive levels against MLLMs, we collected 480 valid sets of test data from human subjects using electronic questionnaires. These 480 participants were from native countries of the six selected languages, with 80 individuals per language. The 1,800 questions of M3GIA are then divided into six complete sub-questionnaires by language, with each individual only responsible for completing the sub-questionnaire corresponding to their native language. We provided more details about human participants in *Appendix A.4* and *F.1*.

Table 1: **The accuracy results on 24 MLLMs regarding each cognitive ability.** The best in **bold** and the second-best underlined. All the numbers are presented in decimal and the full score is 100.

| Types (LLM Size) | Models | ViT Size | Gf | | | | Gc | Gq | Grw | Gv | Overall Acc |
| --- | --- | --- | --- | --- | --- | --- | --- | --- | --- | --- | --- |
| | | | I | RG | RQ | Overall | | | | | |
| **Human** | Average Performance | - | **86.8** | **60.0** | **71.2** | **69.7** | **79.1** | **65.4** | **78.1** | **81.1** | **76.9** |
| API | GPT-4o | - | **58.0** | **59.2** | 33.9 | 50.1 | 72.3 | 42.8 | **79.6** | 46.3 | 59.8 |
| | GPT-4v | - | 56.7 | 56.3 | 40.9 | 51.9 | 74.8 | 46.4 | 77.5 | 52.4 | 59.2 |
| | Gemini-1.5-Pro | - | 54.3 | 56.4 | **41.8** | **54.3** | **75.8** | **60.8** | 77.1 | **53.8** | **62.4** |
| | Gemini-Pro | - | 39.0 | 30.8 | 22.7 | 32.4 | 56.5 | 31.7 | 67.1 | 43.1 | 46.5 |
| | Cluade3-Sonnet | - | 39.7 | 32.9 | 27.3 | 34.0 | 58.3 | 34.2 | 61.3 | 43.9 | 47.0 |
| | Cluade3-Haiku | - | 35.3 | 35.8 | 30.3 | 33.1 | 55.8 | 33.3 | 57.9 | 36.4 | 43.1 |
| OSS (Large) | Mini-Gemini-34b | 0.3B | 37.7 | 37.5 | 30.6 | 34.8 | 61.0 | 34.2 | 62.9 | 45.7 | 48.2 |
| | Mini-Gemini-8*7b | 0.3B | 28.7 | 30.0 | 26.7 | 30.3 | 58.1 | 35.0 | 61.3 | 41.9 | 44.8 |
| | LLaVA-v1.6-34b | 0.3B | 20.7 | **40.0** | 28.5 | 30.8 | 53.8 | 36.4 | 61.7 | 40.4 | 42.8 |
| | Yi-VL-34b | 0.6B | 25.0 | 32.9 | **35.8** | 29.5 | 48.1 | 29.2 | 54.6 | 35.7 | 38.2 |
| | InternVL-chat-v1.2-plus | 6B | **45.0** | **42.5** | 32.4 | **42.5** | **64.6** | **41.4** | **66.7** | **47.5** | **51.9** |
| OSS (Medium) | Mini-Gemini-13b | 0.3B | 22.3 | **29.2** | 23.3 | **24.3** | 41.5 | 26.1 | 44.2 | 28.3 | 32.9 |
| | LLaVA-v1.5-13b | 0.3B | 17.7 | 26.3 | 15.2 | 19.9 | **42.1** | 20.3 | 40.0 | **28.8** | 30.4 |
| | LLaVA-v1.6-vicuna-13b | 0.3B | **23.3** | 19.6 | **24.5** | 23.1 | 36.7 | **26.9** | 47.5 | 28.5 | **33.2** |
| OSS (Small) | Fuyu-8b | - | 21.7 | 22.1 | 27.3 | 23.3 | 27.3 | 24.4 | 27.1 | 24.9 | 25.1 |
| | Mini-Gemini-8b | 0.3B | **37.3** | 29.6 | **31.8** | **30.4** | 51.5 | **30.6** | 56.3 | 36.1 | **41.4** |
| | LLaVA-v1.5-7b | 0.3B | 18.0 | 25.0 | 15.8 | 19.7 | 41.5 | 19.7 | 35.0 | 25.7 | 28.4 |
| | LLaVA-v1.6-vicuna-7b | 0.3B | 21.3 | 22.9 | 18.2 | 20.5 | 36.5 | 19.4 | 32.9 | 26.9 | 31.5 |
| | LLaVA-v1.6-mistral-7b | 0.3B | 24.3 | 25.8 | 24.5 | 24.9 | 38.5 | 24.2 | 36.7 | 32.1 | 28.9 |
| | Deepseek-VL-7b | 0.38B | 32.3 | 29.2 | 22.1 | 28.3 | 50.4 | 24.4 | 54.2 | 32.4 | 37.5 |
| | Yi-VL-6b | 0.6B | 25.2 | **35.5** | 26.2 | 28.8 | 35.6 | 29.0 | 54.5 | 30.8 | 34.4 |
| | Qwen-VL | 1.9B | 18.7 | 23.8 | 25.2 | 22.5 | 41.0 | 27.5 | 42.5 | 30.1 | 32.1 |
| | CogVLM2-LLaMA3-Chinese | 10B | 29.7 | 21.7 | 29.7 | 26.5 | **54.8** | 27.2 | 37.9 | **40.3** | 38.7 |

## 4.1 ACCURACY SCORE ON FIVE COGNITIVE FACTORS

We report the accuracy of each type of question for the 24 models alongside the average human performance for each cognitive ability in Table. 1. We categorize the models into groups by their types, where open-source (OSS) MLLMs are grouped according to the size of their LLMs. It's observed that even the most advanced MLLMs only marginally meet the passing line (60) for overall accuracy, e.g., Gemini-1.5-Pro (62.4) / GPT-4o (59.8) vs human (76.9). Notably, these models excel in domains related to verbal skills and knowledge, such as Gc and Grw. This success can likely be attributed to the powerful language capabilities inherent in large language models, bolstered by their extensive training datasets.

However, a significant gap remains between MLLMs and humans in areas like Visual-Spatial Abilities (Gv) and Fluid Reasoning (Gf). This is particularly evident in the Visual-Spatial Abilities domain, where all models lag considerably behind human capabilities, e.g., Gemini-1.5-Pro (53.8) vs human (81.1). This underscores a substantial opportunity for advancements in the visual aspects of MLLMs. See *Appendix* for case studies. Furthermore, our findings also highlight a pronounced deficiency in the Fluid Reasoning (Gf) capability among all MLLMs, particularly in tasks involving Induction (I) and Quantitative Reasoning (RQ). However, it is surprising to note that in the domain of Deductive Reasoning (RG), the most advanced MLLMs, such as GPT-4o, are approaching the average human level with scores of 59.2 compared to 60.0 for human participants. This might be attributed to the strategy of using synthetic reasoning data to enhance such ability (Chung et al., 2024).

Overall, MLLMs perform well in crystallized intelligence (Gc), possibly owing to their extensive training data, while the most advanced MLLMs still have a large gap with humans in fluid intelligence. This proves that our benchmark M3GIA can measure the difference between crystallized intelligence and fluid intelligence of MLLMs from a cognitive perspective, which is the key difference between M3GIA and other benchmarks.

**Winner Takes All.** More importantly, our finding reveals an intriguing *Winner Takes All* phenomenon that merits further attention beyond the initial observations. Specifically, we noted a

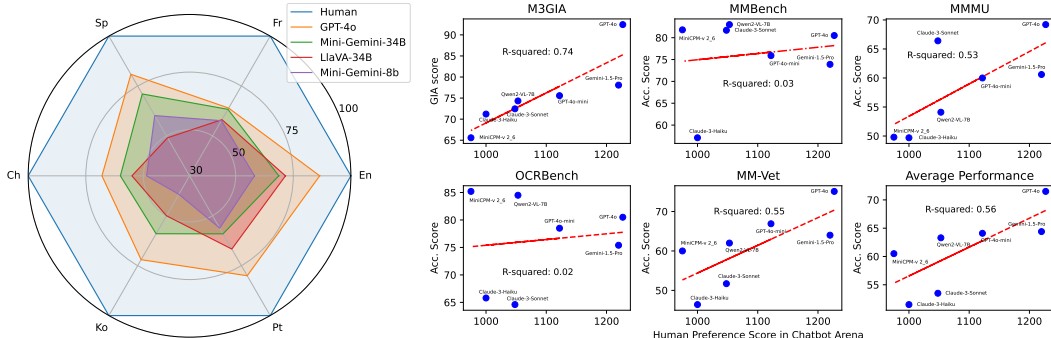

Figure 4: **(Left)** The GIA scores across the six languages. We designate the average human score as 100 and normalize the scores, making the GIA scores comparable across languages. **(Right)** Correlation between models' scores on Chatbot Arena (Vision) (Chiang et al., 2024) and their scores on different benchmarks. M3GIA aligns more effectively with actual human experience.

consistent trend within each group of models where proficiency in one cognitive domain often translates into superior performance across other domains as well in Table 1. In particular, despite the diversity in score distribution among different abilities, there is a noteworthy pattern: the models achieving the top and second-best scores across various cognitive abilities are predominantly the same two models within each group.

This shows an interesting consistency to the pattern observed in human intelligence which empirically suggests the existence of General Intelligence Ability (see Sec. 2). Therefore, it offers compelling evidence that general intelligence ability, also identified as the general factor of intelligence ("g") at the stratum III of the CHC model, has also emerged in large models. Furthermore, it suggests that as MLLMs evolve towards more comprehensive cognitive processes, they too demonstrate a foundational GIA factor that simultaneously governs a variety of cognitive abilities.

## 4.2 MULTILINGUAL GIA SCORES

By collecting a large amount of testing data from human subjects, we adopted CFA (Confirmatory Factor Analysis) model to calculate the GIA scores which can reflect comprehensive intelligence factors. Since the questions for each language are not exactly the same, we need to establish a separate CFA model for each language. We report the GIA scores of each language for some MLLMs in Table. 3, Fig. 4 (left) and Fig. 5. It's observed that the current state-of-the-art MLLM has barely fell within the error bar range of human performance in English. However, these MLLMs still exhibit a significant performance gap compared to humans in other languages.

We perform linear regression to calculate the $R^2$ correlation between models' *human preference score* on Chatbot Arena and their scores on various benchmarks, including MMMU, MMBench (MMB), OCRBench (OCRB) (Liu et al., 2023c), MMVet, and the average performance (Avg.) across 8 prominent benchmarks: MMMU, MMB, HallusionBench (Guan et al., 2023), MMVet, OCRB, AI2D (Kembhavi et al., 2016), MMStar (Chen et al., 2024), MathVista.

Table 2: **M3GIA achieves the best alignment** ($R^2 = 0.74$) with human preference scores on Chatbot Arena among all evaluated benchmarks. This suggests that the GIA score better reflects human preferences and the true capabilities of models compared to traditional task-oriented benchmarks. Detailed results can be found in *Appendix H.2*.

|  | M3GIA | MMB | MMMU | OCRB | MMVet | Avg. |
|---|---|---|---|---|---|---|
| $R^2$ | **0.74** | 0.03 | 0.53 | 0.02 | 0.55 | 0.56 |

**The Impact of LLM Size and Vision Encoder.** We conduct the ablation study with the Qwen1.5 series. (1) Firstly, we trained the models with strictly the same data and use the same ViT component (CLIP-ViT-L-14). As shown in Fig. 5, the GIA scores increase with the rise in LLM size. However, there is often no improvement in cognitive abilities from 7B to 13B, and seems

Table 4: The number of samples seen by the vision encoder has a more significant impact on models' Gv-related performance than the parameter size.

|  | ViT-L/14 | ViT-L/14 | ViT-H/14 | ViT-G/14 |
|---|---|---|---|---|
| Params. | 303M | 303M | 632M | 3000M |
| Samples seen | 13B | 32B | 32B | 34B |
| Gv Acc. | 0.308 | 0.383 | 0.375 | 0.392 |

Table 3: **The General Intelligence Ability of different models accross the six languages.** The left side displays the actual GIA scores, while the right side shows the normalized results after setting the average human GIA scores for each language to 100.0.

| Models | General Intelligence Ability (GIA) | | | | | | Normalized GIA Scores | | | | | |
|---|---|---|---|---|---|---|---|---|---|---|---|---|
| | En | Ch | Fr | Sp | Pt | Ko | En | Ch | Fr | Sp | Pt | Ko |
| Human | 16.01 | 16.69 | 19.52 | 16.22 | 16.00 | 18.05 | 100.0 | 100.0 | 100.0 | 100.0 | 100.0 | 100.0 |
| GPT-4o | 13.85 | 11.46 | 12.37 | 13.12 | 12.80 | 13.01 | 86.5 | 68.7 | 63.3 | 80.9 | 80.0 | 72.1 |
| GPT-4v | 12.61 | 10.95 | 13.83 | 14.04 | 12.12 | 12.25 | 78.8 | 65.6 | 70.8 | 86.5 | 75.8 | 67.9 |
| LLaVA-1.6-34b | 11.47 | 9.25 | 11.35 | 7.96 | 10.67 | 9.04 | 71.6 | 55.4 | 58.1 | 49.1 | 66.7 | 50.1 |
| LLaVA-1.6-13b | 6.96 | 6.89 | 8.71 | 7.75 | 6.94 | 7.75 | 43.5 | 41.3 | 44.6 | 47.8 | 43.4 | 42.9 |
| LLaVA-1.6-7b | 6.75 | 5.99 | 7.67 | 6.74 | 6.01 | 5.93 | 42.1 | 35.9 | 39.3 | 41.5 | 37.6 | 32.9 |
| Mini-Gemini-34b | 11.00 | 9.96 | 12.75 | 11.52 | 9.45 | 10.69 | 68.7 | 59.7 | 65.3 | 71.0 | 59.1 | 59.2 |
| Mini-Gemini-13b | 8.68 | 7.76 | 8.65 | 7.73 | 7.10 | 7.98 | 54.2 | 46.5 | 44.3 | 47.7 | 44.4 | 44.2 |
| Mini-Gemini-8b | 9.32 | 8.11 | 11.25 | 9.76 | 8.99 | 7.08 | 58.2 | 48.6 | 57.6 | 60.2 | 56.2 | 39.3 |
| Qwen-72b[†] | 11.68 | 10.75 | 10.20 | 10.50 | 9.71 | 9.76 | 72.9 | 64.4 | 52.2 | 64.7 | 60.7 | 54.1 |
| Qwen-32b[†] | 10.58 | 9.79 | 9.62 | 10.11 | 9.25 | 9.18 | 66.1 | 58.7 | 49.3 | 62.3 | 57.8 | 50.9 |
| Qwen-14b[†] | 8.46 | 8.76 | 9.15 | 8.49 | 8.79 | 8.32 | 52.8 | 52.5 | 46.9 | 52.4 | 54.9 | 46.1 |
| Qwen-7b[†] | 8.56 | 8.93 | 8.98 | 8.42 | 8.77 | 8.41 | 53.4 | 53.5 | 46.0 | 51.9 | 54.8 | 46.6 |
| Qwen-1.8b[†] | 7.34 | 6.56 | 8.01 | 7.37 | 6.49 | 6.48 | 45.8 | 39.3 | 41.0 | 45.5 | 40.6 | 35.9 |

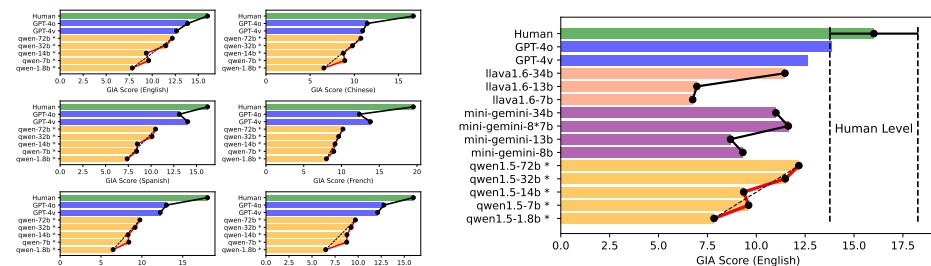

Figure 5: **The GIA scores across the six languages, with Qwen-1.5 LLM series from 1.8B to 72B. (Left)** Generally, the GIA scores increase with the rise of LLM parameters. However, a threshold is observed when scaling up the LLMs' size from 7B to 14B. **(Right)** Taking English as an example, we visualized the performance of various models and compared it with the human level.

to be a emerging point of General Intelligence Ability between 13B and 34B. (2) Furthermore, we trained a series of MLLMs based on Qwen1.5-7B using the same data but with different vision encoders. The results in Table. 4 highlight the importance of data diversity and quantity in enhancing models' Gv ability of effectively encoding visual inputs for downstream tasks.

## 5 CONCLUSION AND LIMITATIONS

**Conclusion.** This paper presented M3GIA, the first "IQ test" that comprehensively evaluate the cognitive abilities of MLLMs under the theoretical umbrella of the well-recognized Cattell-Horn-Carroll (CHC) Model of Intelligence. To meet the pressing need for multilingual assessment, our data spans across six languages and are collected from native sources, including English, Chinese, French, Spanish, Portuguese and Korean. Our results show that the evaluation of M3GIA achieves the best alignment with human preferences compared to traditional task-oriented benchmarks.

**Limitations.** We plan to expand M3GIA to include more rare languages in the future. Unlike normal benchmarks, M3GIA not only involves data collection but also requires significant human effort to create original questions from scratch. This demands a large number of professionals who are native speakers of these rare languages, which introduces considerable costs in both time and funding. Therefore, we plan to prioritize the expansion after the current version is released and recognized. Given the scarcity of multilingual and multimodal benchmarks in the MLLM community, we believe that M3GIA, as the first 'IQ test' for MLLMs, will still make a valuable contribution to the field.

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
