# A  DATASET DOCUMENTATION AND ETHICS STATEMENT

## A.1  MOTIVATION

M3GIA is a multimodal and multilingual benchmark designed to evaluate the cognitive abilities and general intelligence of MLLMs under the theoretical underpinning of human cognition. Instead of leveraging well-developed cognitive science to understand the intelligence of MLLMs beyond superficial achievements, existing benchmarks still mainly focus on evaluating solely on task performance. As described in the paper, these approaches have several limitations. We aim to bridge this gap through M3GIA, providing helpful insights into the development of artificial intelligence models with true intelligence.

## A.2  MORE DETAILS ON DATA COLLECTION AND ANNOTATION

**Source of the data:**    Since human IQ tests are not open to the public, and considering the novelty of some question types (logo problem, comic problem, etc.), 73% of our data are crafted by ourselves. Images come from two sources: (1) For human cognitive tasks like Visualization and Concept Formation, we manually created figures using PowerPoint or Visio; (2) For tasks like comics, we collected materials from public websites. We trained annotators to follow three principles: (i). Image clarity; (ii). Avoid taking screenshots of images that are prohibited from being downloaded to prevent copyright infringement; (iii). Collect data from websites in the corresponding language. It's important to note that while the images are sourced from public resources, the questions posed about the materials are original, thereby avoiding potential data leakage. Another 27% of data are collected from existing sources. For this part of data, we follow the practice in MMMU (Yue et al., 2023) and M3Exam (Zhang et al., 2024), whose data are collected from existing human tests.

**How we crafted the questions:**    We use a two-phase approach for self-designed questions (creation-review). For example, for tasks like Visualization and Number Series, we hired people with psychology background to create the questions. Most images were created by our team using PowerPoint or Visio (depending on the annotator's preference). Once completed, the questions undergo the review process (See Appendix D). Questions based on existing materials are crafted in three-phases. For instance, after collecting an original article from the web, annotators will create reading questions based on the article, followed by peer review. For the remaining 27%, since the collected questions were mainly in documents, we need to undergo meticulous processing. This include: OCR for editable versions, converting formulas to LaTeX, removing gibberish, and so on. Our annotators are professional and are recruited from *** company, which provides reliable human annotated datasets for some of the world's biggest brands (we hide the company name to meet the requirements for anonymity). The annotators have at least bachelor's degree, and are native speakers.

## A.3  HOW THE FIVE FACTORS ARE CHOSEN FOR EVALUATING MLLMS?

The most mainstream version of the CHC model includes nine cognitive factors: However, most widely recognized CHC-based human IQ tests, do not typically incorporate all nine factors (Roid & Pomplun, 2012; Schrank & Wendling, 2018). For example, the Stanford-Binet Intelligence Scales select 5 factors (Gc, Gf, Gq, Gv, Gwm), while the WJ IV Tests of Cognitive Abilities select 7 factors (Gc, Gf, Gv, Gwm, Gs, Glr, Ga). In other words, it is not necessary to include all factors for a test to be effective. Here, we listed the reasons for excluding the 4 factors one by one:

- **Memory Factors (Gsm, Glr):** The concept of memory in machine intelligence differ from that in human. Memory typically involves the retention and forgetting of information over time (Baddeley, 1992; Spear, 2014). However, present MLLMs cannot think continuously over time; their "thinking" is discrete rather than continuous. Whether 5 seconds or 15 minutes pass before starting the next round of dialogue makes no difference to the model, meaning it cannot measure "the degree to which information is forgotten over time." What appears similar to human memory in these models is multi-turn dialogue, where the model is asked if it still "remembers" information after multiple rounds of dialogue. However, if we take a closer look to the operating mechanism of LLMs, previous dialogue is fixly stored in the computer's memory as context and directly used as explicit input in the next Q&A session (similar to humans writing down all previous conversations on a whiteboard and being able to see that information explicitly when speaking next). Given the

significant difference in mechanisms, comparisons with human baselines are difficult and lack significance, making it challenging to construct GIA calculation models based on human data.

- **Processing Speed (Gs):** Human reaction speed is a critical factor in assessing intelligence, as it is directly related to the sensitivity of sensory organs and the functional state of the central nervous system (Deary et al., 2010). In contrast, the processing speed of MLLMs depends on factors such as hardware or the use of acceleration technics which are external factors unrelated to the model itself, it becomes challenging to control variables consistently when different users measure performance using our benchmark.

- **Auditory Processing (Ga):** Our M3GIA primarily targets vision-language models and does not yet include audio modalities. Although some of the latest multimodal models are beginning to incorporate audio, models capable of handling images, text, and audio simultaneously are still rare. Therefore, this version of M3GIA does not include the audio modality.

Besides, in some other versions of the CHC model, certain sub-abilities originally classified under stratum II have been elevated to independent CHC factors. However, (i) These newly added factors are still under discussion and have not been empirically validated. (ii) The factors include: Olfactory Abilities (Go), Tactile Abilities (Gh), Kinesthetic Abilities (Gk), Psychomotor Abilities (Gp), Psychomotor Speed (Gps), Reaction Speed (Gt), and Domain Specific Knowledge (Gkn). Gk, Gp, Gps, and Gt are primarily related to physical functions and motor abilities, while Go and Gh involve olfactory and tactile modalities, which are not currently addressed by multimodal large models. On one hand, the MLLMs targeted by M3GIA are not yet equipped to measure these factors. On the other hand, even in current human IQ tests, these factors are not typically considered, as they represent broader capabilities rather than a narrow focus on intelligence. We will consider incorporating these factors in the future as MLLMs evolve. This also highlights one of the reasons for choosing CHC as the theoretical framework: it leaves space for future expansion of our benchmark under the same theoretical umbrella as MLLMs evolve.

**Psychological Support** What is more, the rationale for selecting the five factors is also well-supported by psychological validation factor analysis (Phelps et al., 2005). In the study, the factors with the highest significant factor loadings in relation to general intelligence are as follows: Gf (0.98), Gq (0.87), Glr (0.84), Gc (0.79), Gsm (0.78), and Gv (0.68), while Ga and Gs only have loadings of 0.47 and 0.48, respectively. Among these, the factors with higher significant loadings have been included in our considerations, except for Glr and Gsm, which were excluded for the reasons outlined above.

A.4 How Human Participants were Selected and Tested

Participants were recruited via two platforms: the NAODAO Psychology online testing platform for Chinese participants, and Amazon Mechanical Turk for participants of other languages. Participation was compensated and entirely voluntary. Individuals were provided with a link. After giving informed consent, interested participants were directed to the anonymous online battery of questionnaires. To participate, individuals should be native speakers of the language they were tested in, within 22 and 35 years old (Elam et al., 2021). Within these criteria, participant selection was random. To motivate thoughtful responses, the compensation was structured incrementally (Up to $50).

The average duration is 5.5 hours (maximum limit of 8 hours). To ensure quality, we randomly inserted "check questions". For instance, a check question might instruct participants to "Please select option B." If a participant answered more than 2 such questions incorrectly, their submission would be considered invalid. The study was reviewed and approved by the Technology Ethics Governance Committee of <XXX> Group (only write <XXX> to meet the requirements for anonymity).

A.5 Composition

- M3GIA contains a total of 1.8K multiple-choice problems. We ensure that all VQA tasks necessitate reliance on images for resolution and cannot be resolved with text alone (see Sec. E.2). M3GIA includes question sets in six languages, comprising Chinese, English, Spanish, Korean, Portuguese, and French.

- Each question is labeled with one or several CHC factors, with involved factors marked as '1' and non-involved factors marked as '0'. Each question is also annotated with the question cluster and the narrow question type to which it belongs, to facilitate the calculation of accuracy rates.

- M3GIA is self-contained. We bear all responsibility in case of violation of rights.

- The dataset does not contain any information that might be offensive, insulting, or threatening.

## A.6 Usage and Distribution

- The dataset is released anonymously at https://anonymous.4open.science/r/M3GIA-v1-23E3.

- The data is saved in Parquet format, where an example is shown in the README.md file. An example code snippet is also provided showing how to read and process the data.

## A.7 Maintenance

- M3GIA will be managed and maintained by our research group.

- If we further expand our dataset or find any errors, we will update the dataset and results in the leaderboard accordingly. It will be updated on our website (not publicly disclosed yet due to anonymity requirements).

## B Definitions of the CHC factors

According to the Cattell-Horn-Carroll (CHC) Model of Intelligence(Schneider & McGrew, 2012; 2018), the definitions of the five cognitive factors are as follows:

**Comprehension-Knowledge (Gc)**, also known as *Crystallized Intelligence*, is the knowledge of culture that is incorporated by individuals through a process of "acculturation" (McGrew, 2009). Gc is typically described as the breadth and depth of acquired knowledge of the language, information and concepts of a culture, and the application of the knowledge. Gc is primarily a store of verbal or language-based declarative (knowing what) and procedural (knowing how) knowledge acquired during general life experiences. In short, Gc reflects the ability to apply and reason using previously learned experiences and common knowledge. (Schneider & McGrew, 2012)

**Fluid Reasoning (Gf)** is the broad ability involved in reasoning, forming concepts, and solving problems using unfamiliar information or in novel situations. It includes inductive, deductive, and quantitative reasoning and *is typically evident in mental operations, such as inferential reasoning, forming concepts, classification of unfamiliar stimuli and recognizing patterns.* (McGrew, 2009; Schneider & McGrew, 2012) Furthermore, there are three factors that are generally considered the hallmark indicators of Gf:

- **Induction (I).** The ability to observe a phenomenon and discover the underlying principles or rules that determine its behavior.

- **Deductive Reasoning (RG).** This ability, also known as general sequential reasoning, refers to the capacity to reason logically using known premises and principles step by step.

- **Quantitative Reasoning (RQ).** The ability to reason, either with induction or deduction, with numbers, mathematical relations, and operators.

**Visual-spatial Processing (Gv)** is the ability to perceive, analyze, synthesize, and think with visual patterns, or more succinctly, "the ability to make use of simulated mental imagery to solve problems". Once the eyes have transmitted visual information, the visual system of the brain automatically performs a large number of low-level computations (e.g., edge detection, light/dark perception, color-differentiation, motion-detection, and so forth). The results of these low-level computations are used by various higher-order processors to infer more complex aspects of the visual image. (Schneider & McGrew, 2012). Gv abilities are typically measured by tasks (figural or geometric stimuli) that require the perception and transformation of visual shapes, forms, or images and/or tasks that require maintaining spatial orientation with regard to objects that may change or move through space. (McGrew, 2009)

Table 1: **The number and cognitive factors of each question type.** Our M3GIA is organized into five clusters, each cluster is further defined to combine two or more narrow question types that are aspects of a broad CHC construct (real-world problems in **bold**). In total, it contains a total of 1,800 meticulously designed multilingual questions, with the number of questions and the distribution of question types being completely consistent across different languages. Questions potentially related to the cultural backgrounds are marked in green, while purely intellectual questions, unrelated to cultural background, are marked in yellow. The former's data are sourced from native language context, while the latter uses questions translated into the six languages.

| Cluster | Question Types | Gc | Gv | Grw | Gq | Gf I | Gf RG | Gf RQ | Num |
|---|---|---|---|---|---|---|---|---|---|
| Common Sense | General Information | ✓ | | | | | | | 20 × 6 |
| | Oral Vocabulary | ✓ | | | | | | | 15 × 6 |
| | **Logo Problem** | ✓ | ✓ | | | | | | 15 × 6 |
| Visual-spatial | Visualization | | ✓ | | | | | | 30 × 6 |
| | Picture Recognition | | ✓ | | | | | | 15 × 6 |
| | **Real-world Spatial** | | ✓ | | | | | | 15 × 6 |
| Comprehension | Readings-text | | | ✓ | | | | | 15 × 6 |
| | Readings-VL | | | ✓ | | | | | 10 × 6 |
| | **Comic Problem** | | ✓ | ✓ | | | | | 15 × 6 |
| Mathematics | Math Facts | | | | ✓ | | | | 25 × 6 |
| | Algebra | | | | ✓ | | | ✓ | 15 × 6 |
| | Geometry | | ✓ | | ✓ | | | ✓ | 10 × 6 |
| | **Applied Problem** | ✓ | | | ✓ | | | ✓ | 10 × 6 |
| Reasoning | Number Series | | | | | ✓ | | ✓ | 20 × 6 |
| | Concept Formation | | | | | ✓ | | | 20 × 6 |
| | Raven's Matrices | | ✓ | | | ✓ | | | 10 × 6 |
| | Syllogism Problem | | | | | | ✓ | | 20 × 6 |
| | **Real-world Reasoning** | ✓ | | | | | ✓ | | 20 × 6 |

**Reading and Writing (Grw)** is the depth and breadth of knowledge and skills related to written language. It is worth noting that, although reading and writing are clearly distinct activities, the underlying sources of individual differences in reading and writing skills do not differentiate between the two activities cleanly (Schneider & McGrew, 2012). It appears that the ability that is common across all reading skills also unites all writing skills.

**Quantitative Knowledge (Gq)** is the depth and breadth of knowledge related to mathematics. Specifically, it is the ability to comprehend quantitative concepts and relationships and to manipulate numerical symbols. It consists of acquired knowledge about mathematics such as knowledge of mathematical symbols (e.g., $\int, \pi, \sum, \infty, \neq, \leq, +, -, \times, \div$, and many others), operations (e.g., addition/subtraction, multiplication/division, exponentiation/nth rooting, factorials, negation, and many others), computational procedures (e.g., long division, reducing fractions, quadratic formula, and many others). Gq abilities are typically measured by tests include measures of math calculation, applied problems (or math problem solving), and general math knowledge (e.g., Arithmetic on the Wechsler Scales, Quantitative Reasoning on the SB5).

## C  INTRODUCTION TO THE EVALUATION QUESTIONS

In this section, we will outline the five question clusters and the 18 narrow question types they encompass.

**The Common Sense Cluster.** The common sense cluster is designed to measures the Gc factor of an MLLM and includes 3 narrow question types: general information, oral vocabulary and logo problem. In **general information**, the model is presented with an image and is asked, "Where would you find [the object] in the picture?" or "What would you do with [the object] in the picture?" The initial items in each subtest draw from familiar everyday objects, and the items become increasingly difficult as the objects become more obscure or less familiar. **Oral vocabulary** consists of two subtests: Synonyms and Antonyms. In the Synonyms subtest, the model is provided with a word and is asked to choose its synonym. In the Antonyms subtest, the model is provided with a word and is asked to choose its antonym. In CHC theory, this test primarily measures a narrow aspect of Comprehension-Knowledge (Gc) referred to as lexical knowledge (VL; vocabulary knowledge), or knowledge of words and word meanings. (Schrank et al., 2016) The **logo problem** is the real-world problem of the cluster, where a model is provided with a logo and is required to identify an abstract element within it. To achieve this, it must have a very deep impression on the element, such as a confusing artistic characters or symbolic expression of cultural elements, which requires a high level of Gc and a certain level of Gv.

**The Visual-spatial Cluster.** This cluster is designed to evaluate the Gv factor and includes 3 narrow question types: visualization, picture recognition and real-world spatial. **Visualization** consists of two subtests: Block Rotation and Spatial Relations. In the former, the model is asked to identify the rotated 3D block that match the original 3D block. In the latter, the model is required to identify three or four pieces that form a complete target shape. In **picture recognition**, a model is asked to identify a subset of specified pictures within a field of distracting pictures. The stimuli and distracters for each item include varieties of the same type of object (e.g., several different leaves) to eliminate verbal mediation as a memory strategy (Schrank & Wendling, 2018). **Real-world spatial problem** necessitates that the model accurately determines the relative 3D positioning of objects within an image depicting real-world scenarios. This requires the model to recognize and interpret all existing relationships in the physical world, including comprehensive 3D spatial relationships and the dynamic interconnections between the objects portrayed.

**The Comprehension Cluster.** This cluster is designed to evaluate the Grw factor and includes 3 narrow question types: readings-text, readings-VL and the comic problem. In **readings-text**, the model is provided with long articles (about 4-6 paragraphs) and will be required to answer questions related to the main ideas of the articles or the relationships between paragraphs. The articles are collected from reading comprehension exercises found in middle and high school levels across the six countries. To highlight the multimodal nature of our benchmark, we designed **readings-VL**, where responses must be selected from image-based options besides the conventional text-based queries. In the **comic problem**, the model will be provided with a comic consisting of four or more panels that make up a complete plot. To answer the questions, the model needs to understand the entire story's connotation based on the textual dialogues between characters and the plot development. This approach evaluates the model's ability to integrate visual narrative comprehension with textual comprehension, challenging it to understand scenarios represented both visually and textually.

**The Mathematics Cluster.** This cluster is designed to evaluate the Gq factor and includes 4 narrow question types. **Math facts** is tailored to measure Gq alone and consists of two subtests: symbolic knowledge and geometric knowledge. The former focuses on the model's acquired knowledge about mathematical symbols and operations. It covers knowledge from elementary to university level, including arithmetic, vector operations, calculus, etc. The latter emphasizes the model's capability to solve problems using geometric knowledge. In **algebra** and **geometry**, we source the questions from authentic middle school and high school exam papers across the six countries. Unlike math facts problem which can be directly answered once the knowledge is acquired, these problems require a further reasoning process. Thus, they not only call upon Gq but also require RQ. To evaluate the model's ability to solve mathematical problems in real-life scenarios, we have specially designed **application problems**. For example, the model might be provided with a restaurant bill and asked to calculate the total amount to be paid. Since it rely heavily on common knowledge, Gc is also annotated in this type of problems.

**The Reasoning Cluster.** This cluster is designed to assess the Gf factor and includes five narrow question types. Specifically, **number series**, **concept formation**, and **Raven's Matrices** are targeted at evaluating the I (inductive) factor, while the **syllogism problem** and **real-world reasoning** target the RG (deductive reasoning) factor. In **number series**, the model is presented a series of

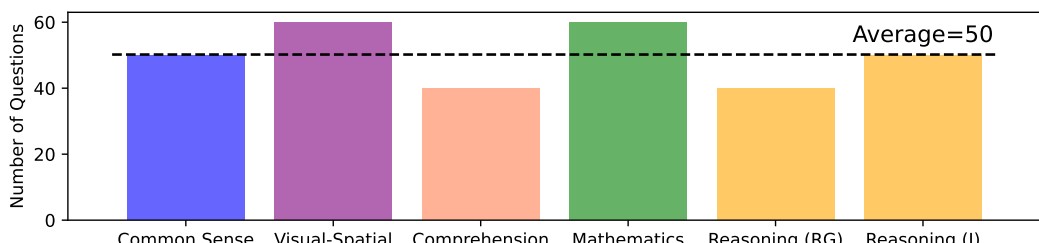

Figure 1: **Data Balancing.** we keep the number of questions for each cluster as balanced as possible when collecting questions. Given the unique characteristics of Gf, we have divided the Reasoning Cluster into Reasoning (I) and Reasoning (RG) for statistical analysis. Across each language, the number of questions within each cluster varies from 40 to 60, with an average of 50.

numbers with one or more numbers missing. The model must determine the numerical pattern and provide the missing number in the series. **Concept formation** measures the ability to categorize and compare (Andrewes, 2015), a basis for abstracting concepts (Wang, 2019). It requires the model to examine a series of shapes or pictures and then formulate a rule that applies to the item and then figure out the item that do not coincide with the rule. The **syllogism problem** is a classic form of deductive reasoning, where the model is presented with two statements followed by two conclusions. The model have to take the statements to be true even if they appear to contradict commonly known facts. Then it is asked to decide which of the given conclusions logically follows from the two given statements, disregarding commonly known facts. **Real-world reasoning** refers to logical reasoning questions rooted in real-world scenarios, where Gc is also important.

## D    CONNECTION BETWEEN THE QUESTION TYPES AND THE CHC FACTORS

To ensure that M3GIA maintains professionalism as a cognitive science test, most of M3GIA's questions adhered to the question designs of the well-recognized WJ-IV (Schrank & Wendling, 2018) for each CHC factor, while the remaining questions were self-designed.

As introduced in sec. 3.1 of the main paper, each CHC factor comprises multiple narrow abilities (sub-factors, also known as Stratum I). For the first part of the questions, each question type is specifically crafted according to the definition of a specific sub-factor within a CHC factor. The content of these question types aligns so closely with the corresponding CHC factor definitions that their selection to assess these factors feels intuitive (see Table. 2). For the latter, each type is crafted to assess a broad CHC factor, according to its definition (see Table. 3).

## E    DATA CURATION PROCESS

### E.1    DATA COLLECTION AND STATISTICS

**Data Balancing.**    To ensure equal consideration for each CHC factor during the assessment, we have maintained a balanced number of questions for each cluster that measures the various CHC factors, as shown in Fig. 1. Specifically, the number of questions in each cluster fluctuates around 50, with a maximum capped at 60 and a minimum threshold of 40.

**Questions Crafted from Scratch.**    Due to the fact that many human intelligence tests are not open to the public, and considering the novelty of some of our question types (such as logo problem, comic problem, etc.), we could not source pre-existing QA pairs from available datasets for many questions. Consequently, we have crafted numerous questions from scratch. For these questions, ensuring the correctness of the answers and the clarity of the descriptions is particularly important. See later Sec. E.2 for more detailed information.

**English-centric Bias.**    Apart from questions that are completely independent of cultural background, such as Number Series and Raven's Matrices, all data are sourced from native websites corresponding to the language. These data encompass not only text explicitly linked to cultural backgrounds but

Table 2: **The close connection between the question types and the CHC factors.** These question types adhered to the question designs of existing cognitive tests, including the well-recognized WJ-IV and Raven. As introduced in Sec. 3.1 of the main paper, each CHC factor comprises multiple narrow abilities (sub-factors, also known as Stratum I). Each question type is meticulously designed based on the definition of a specific sub-factor within a CHC factor (Schrank et al., 2016).

| Question Types | CHC (sub-factor) Definition | Content of the question |
|---|---|---|
| General Information | Gc (K0): The store of language-based or verbal declarative (knowing what) and procedural (knowing how) knowledge acquired during general life experiences. | The model is presented with an image and is asked, "Where would you find [the object] in the picture?" or "What would you do with [the object]?" |
| Oral Vocabulary | Gc (VL): Knowledge of the definitions of words and the concepts underlie them. | The model is provided with a word and is asked to choose its synonym or antonym. |
| Visualization | Gv (Vz): The ability to perceive complex patterns and mentally simulate how they might look when transformed (e.g., rotated, changed in size, partially obscured, and so forth). | It consists of two subtests: In Block Rotation, the model is asked to identify the rotated 3D block that match the original 3D block. In Spatial Relations, the model is required to identify pieces that form a complete target shape. |
| Picture Recognition | Gv (MV): The ability to remember and identify complex images, also known as Visual Memory. | The model is presented with a shape, and is asked to identify the shape within a field of distracting shapes. |
| Readings-text Reading-VL | Grw (RC): The ability to understand written discourse. | The model is required to answer questions related to the main ideas of long articles (4-6 paragraphs) or the relationships between paragraphs. |
| Math Facts | Gq (KM): Range of general knowledge about mathematics. This factor is about "what" rather than "how" knowledge. | The questions focuses on the model's acquired knowledge about symbol and geometry, covering from elementary to university level. It doesn't rely on using mathematical knowledge for complex reasoning, but rather focus on the knowledge itself. |
| Algebra Geometry | Gq (A3): Measured (tested) mathematics achievement. The full name of A3 is Mathematical Achievement. | Unlike math facts problem which can be directly answered once the knowledge is acquired, these problems require a further reasoning process. We source the questions from authentic exam papers across the six countries to measure the Mathematical Achievement factor. |
| Number Series | Gf (RQ): The ability to reason, either with induction or deduction, with numbers, mathematical relations, and operators. | The model is presented a numbers series with one or more numbers missing. The model must determine the numerical pattern and provide the missing number. |
| Concept Formation | Gf (I): The ability to observe a phenomenon and discover the underlying principles or rules that determine its behavior. | It requires the model to examine a series of shapes or pictures and then formulate a rule, and then figure out the item that do not coincide with the rule. |
| Raven's Matrices | Gf (I): See above. | The model is asked to identify the missing element that completes a pattern. Patterns are presented in the form of a $4 \times 4$ or $3 \times 3$ matrix. |
| Syllogism Problem | Gf (RG): The ability to reason logically using known premises and principles. This ability is also known as deductive reasoning or sequential reasoning. | It is a classic form of deductive reasoning, where the model is asked to decide which of the given conclusions logically follows from the two given statements. |

Table 3: **The close connection between the question types and the CHC factors.** This part of question types are self-designed questions. For these types of questions, each type is crafted to assess a broad CHC factor, according to its definition.

| Question Types | Precise definition of the CHC factors | Content of the question |
| --- | --- | --- |
| Logo Problem | **Comprehension-Knowledge (Gc)**: The breadth and depth of acquired knowledge of culture that is incorporated during general life experiences (McGrew, 2009). It reflects the ability to apply previously learned experiences and common knowledge (Schneider & McGrew, 2012). | The model is required to identify an abstract element within a logo. To achieve this, it must have a very deep impression on the element, such as a confusing artistic characters or symbolic expression of cultural elements, which requires a high level of Gc (general life knowledge). |
| Real-world Spatial | **Visual-spatial Processing (Gv)**: The ability to perceive visual stimuli and perform spatial imagination (Schneider & McGrew, 2012). | It requires the model to accurately determine the relative 3D positioning of objects within an image depicting real-world scenarios, including all existin 3D spatial relationships in the physical world and the dynamic interconnections between the objects portrayed. |
| Comic Problem | **Reading and Writing (Grw)**: The depth and breadth of knowledge and skills related to written language. People with high Grw read with little effort. Although reading and writing are distinct activities, the underlying sources of individual differences in reading and writing do not differentiate between the two activities (Schneider & McGrew, 2012). | The model is provided with a comic consisting of four or more panels that make up a complete plot. To answer the questions, the model needs to understand the entire story's connotation based on the textual dialogues between characters. |
| Applied Math Problem | **Quantitative Knowledge (Gq)**: The depth and breadth of knowledge about mathematics such as symbols $(\int, \pi, \sum, \infty, \neq, \leq, +, -, \times, \div)$, operations, computational procedures (e.g., reducing fractions, quadratic formula). It is the ability to comprehend quantitative concepts and to manipulate numerical symbols. Gq is typically measured by tests include measures of math calculation, applied problems (or math problem solving) (Schneider & McGrew, 2012) (e.g., Arithmetic on the Wechsler Scales). | Applied math problems are designed to evaluate the model's ability to solve mathematical problems in real-life scenarios. For example, the model might be provided with a restaurant bill and asked to calculate the total amount to be paid. |
| Real-world Reasoning | **Fluid Reasoning (Gf)**: The broad ability involved in reasoning, forming concepts, and solving problems using unfamiliar information or in novel situations. It includes inductive, deductive, and quantitative reasoning and is typically evident in mental operations, such as inferential reasoning, forming concepts, classification of unfamiliar stimuli and recognizing patterns (McGrew, 2009; Schneider & McGrew, 2012). | Real-world reasoning problems refer to complex logical reasoning problems rooted in real-life scenarios, which may require the use of various Gf abilities such as deductive reasoning (RG), induction (I), and quantitative reasoning (RQ). |

also images, since images can also convey information about the cultural contexts implicitly, such as the attire of people in the image background, architectural styles specific to a region, etc.

**Multimodal Nature.** As a multimodal benchmark, safeguarding the dataset's multimodal attributes is crucial. In particular, questions related to images should require the visual information for resolution and not be solvable through text alone. This principle was rigorously adhered to during the data

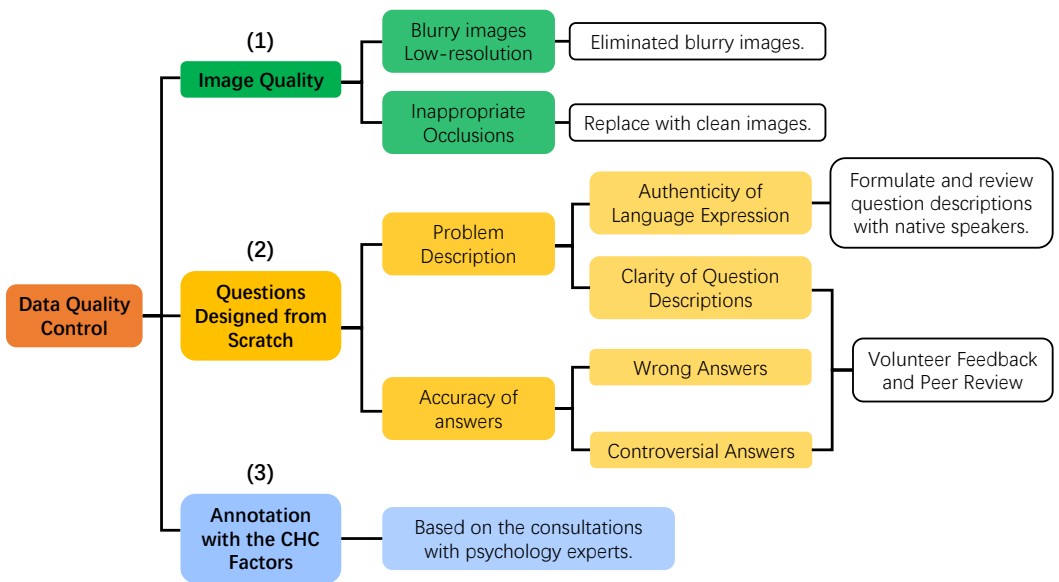

Figure 2: **How we ensure the data quality of M3GIA.**

collection phase, and we also placed emphasis on it during the checking process (see later Sec. E.2). We further validated the importance of image information in our benchmark through an experiment that involved removing images from the evaluation dataset, as shown in Fig. 4.

Table 4: **Comparison of GPT-4v's accuracy rates** across five clusters **before and after the exclusion of images** from the evaluation dataset. Removing images from the dataset resulted in a notable decline in evaluation performance, underscoring the significance of visual information in the assessment and emphasizing the multimodal nature of our M3GIA benchmark.

|  | Common Sense | Visual-Spatial | Comprehension | Mathematics | Reasoning | Overall |
|---|---|---|---|---|---|---|
| With Images | 87.0 % | 48.0 % | 77.8 % | 46.4 % | 56.5 % | 60.7 % |
| Without Images | 44.5 % ($\downarrow$) | 23.9 % ($\downarrow$) | 50.4 % ($\downarrow$) | 31.7 % ($\downarrow$) | 37.8 % ($\downarrow$) | 36.6 % ($\downarrow$) |

E.2   DATA QUALITY CONTROL

To further control the quality of our data, we perform the data cleaning process from four perspectives, as illustrated in Fig. 2.

- **Image Quality.** We traverse the dataset and locate all blurry images with resolutions lower than $100\times100$ px. For questions featuring these images, we either replace them with similar questions that use high-resolution images or substitute the images with clear alternatives that convey the same meaning.

- **Accuracy Check.** For the questions we designed from scratch, we have paid special attention to ensuring their correctness.

  (i) To guarantee the authenticity of the language expression in our questions, we engaged native speakers to both formulate and review the descriptions of the question stems. Specifically, after establishing the intended meaning and creating a draft version, these native speakers undertake a thorough review, culminating in the finalized version of the question descriptions.

  (ii) We employed volunteer feedback and peer review as methods to assess the clarity of our question descriptions and to detect any potential issues with the answers.

  *Clarity of Descriptions:* We recruited 10 volunteers for each language who were not involved in question creation to take our tests and provide feedback on any errors or unclear descriptions

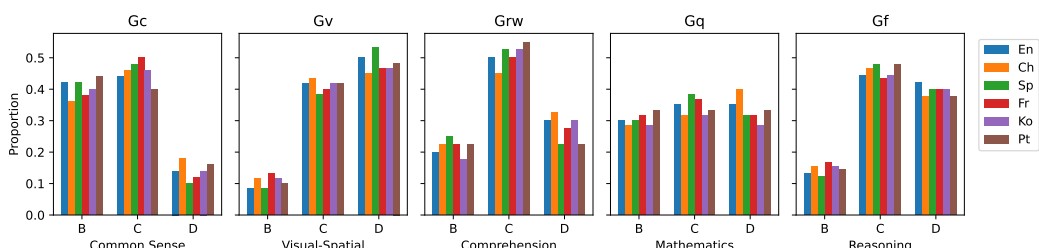

Figure 3: **Difficulty control across languages.** The distribution of difficulty levels across languages is nearly identical.

they encountered in the questions. After thorough discussion of their feedback, we ultimately incorporated revisions into 28 questions.

*Correctness of Answers:* After the volunteers submit their answers to the electronic questionnaire, the correct answers will be automatically disclosed. They will then be prompted to revisit any questions they answered incorrectly and are encouraged to challenge these, offering feedback on any they assert to be correct or view as contentious. This feedback was taken seriously, and we ultimately made corrections to six instances where we recognized that the answers were indeed controversial or misleading. Besides, we also employed peer review within our group to ensure the correctness of answers. Specifically, after formulating their questions, team members will swap them with each other for a round of testing. Following this exercise, if a tester has a justifiable reason for an incorrect response, they will engage in a direct discussion with the question's author. This method led to the identification of around ten answers that were deemed contentious.

- **Annotation of the CHC Factors.** To ensure the rationality of the questions designed for each CHC factor and the validity of the CHC factors annotated for each question, psychologists were deeply involved and cooperated in the question design and annotation phases.

- **Difficulty Control Across Languages.** After each annotator created questions, the questions were tested by three additional annotators in that language, who were not provided with the answers and were asked to rate each question's difficulty into 1 of 5 difficulty levels: A (very easy) to E (very difficult). We filtered out questions that are consistently rated as too easy (A) or too hard (E) and maintained consistency in the number of B, C, D-level questions across languages. Following this initial screening, the questions were reviewed by the psychology expert of our team. The expert further excluded questions deemed too easy or difficult and adjusted the proportions of B, C, D-level questions as necessary. As a result, the distribution of difficulty levels across languages is nearly identical, as shown in Fig. 3.

## F   THE GIA METRICS

### F.1   HUMAN DATA COLLECTION

We collected human data in each language from 80 participants using paid electronic questionnaires. Participation in the test is compensated and entirely voluntary. To protect user privacy, the test is also conducted anonymously. Each participant was mandated to answer all questions to be eligible for payment. To motivate participants to provide thoughtful responses, the compensation is structured incrementally, increasing with the number of questions answered accurately. Additionally, to mitigate the risk of participants choosing answers at random just for the monetary incentive, we randomly inserted several "check question" within the questionnaire. For instance, a check question might instruct participants to "Please select option B." If a participant answer more than two such questions incorrectly, their submission would be considered invalid.

### F.2   CALCULATION OF THE GIA SCORE

In this study, we employed a cognitive factor analysis (CFA) approach to model the General Intelligence Ability (GIA) of human subjects based on the CHC theory of cognitive abilities (Dubois et al.,

2018). The CHC theory posits a hierarchical structure of cognitive abilities, encompassing broad factors such as Gc, Gf, Gv, Gq, and Grw, which are further broken down into narrower tasks. Our MarcoBench, a comprehensive set of 1,800 multiple choice problems corresponding to the assessment of the five CHC cognitive factors, was meticulously subdivided into 18 distinct question types, each designed to measure different facets of the cognitive abilities being assessed.

Data collection involved 80 human subjects across four different languages: Chinese, English, Portuguese, and Korean. A total of 60 subjects were utilized for model building, while the remaining 20 subjects were reserved for model validation. Subjects were administered the MarcoBench, and their performance on the tasks was meticulously recorded. The data comprised accuracy scores on 18 cognitive tasks, representing the 18 distinct question types. The accuracy data was firstly normalized to generate z-scores. And then, the EFAtools package was employed to scale the data and calculate the correlations between the variables. A series of statistical tests, including Bartlett's test and the Kaiser-Meyer-Olkin (KMO) measure, were conducted to assess the suitability of the data for factor analysis. An overall KMO value larger than 0.6 was deemed acceptable for factor analysis (Watkins, 2018).

The CFA model was constructed in accordance with the CHC theory, with the broad and narrow factors defined as per the theoretical framework. We used the lavaan package (https://www.lavaan.ugent.be/) to fit the CFA model to the pre-processed data. The CFA model structure included:

- Gc: Measured through general information, oral vocabulary, and logo problem tasks.
- Gv: Included visualization, picture recognition, and real-world spatial tasks.
- Grw: Assessed through readings-text, readings-visual-language (VL), and comic problem.
- Gq: Comprised math facts, algebra, geometry, and application problems.
- Gf: Evaluated through number series, concept formation, Raven's Matrices, syllogism problem, and real-world reasoning tasks.

Additionally, a General Intelligence Ability (GIA) factor was included, integrating all five broad factors. Model estimation was performed using Maximum Likelihood with Restricted Maximum Likelihood (MLR) estimation, which has been demonstrated to be more robust in the presence of multicollinearity.

The model's fit was evaluated using a range of indices, including the chi-square statistic, degrees of freedom, p-value, Comparative Fit Index (CFI), Root Mean Square Error of Approximation (RMSEA), Standardized Root Mean Square Residual (SRMR), and Akaike Information Criterion (AIC). The primary focus was on the CFI and SRMR, as they are considered more reliable indicators of model fit. A CFI larger than 0.8 or 0.9 was considered acceptable, while an SRMR equal to or lower than 0.08 was deemed acceptable (Baumgartner & Homburg, 1996; Doll et al., 1994).

Upon establishing a satisfactory model fit, we employed it to calculate latent scores for the GIA on a separate set of test data. Subsequently, we calculated the Pearson correlation coefficient between the GIA latent score and the overall accuracy of the subjects on the test data to validate the model's effectiveness. The results of this analysis provided robust evidence for the validity of the CFA model in capturing the GIA of human subjects, as indicated by the significant positive correlation between the GIA latent score and overall accuracy. This validation process underscores the model's theoretical grounding in the CHC theory and its empirical support from the data. Subsequently, we applied the CFA model to estimate the GIA for several MLLMs, including gpt-4o (OpenAI, 2024), gpt-4v (Achiam et al., 2023), llava1.6-34b (Liu et al., 2023a), llava1.6-13b, llava1.6-7b, mini-gemini-34b (Li et al., 2024), mini-gemini-7*8b, mini-gemini-13b, and mini-gemini-8b, enabling a comparative analysis of their cognitive abilities against human performance.

## G   EVALUATION STRATEGY

**Option Extraction**   For choice extraction, we adopted a two-stage strategy. In the first stage, we employed a keyword-based rule method to parse the model output in order to obtain options. This approach proved very effective, with the majority of existing multimodal large models successfully identifying correct answers at this stage. Yet, to enhance the robustness of our evaluation, we adopted a second stage of precautionary measures in case the parsing in the first stage fails. This involves

deploying GPT-4-turbo for the concise summary of answer choices from the original model responses. If the second stage still fails, we will randomly generate an option for the model as the answer to the question. It is noteworthy, though, that throughout the actual testing process thus far, we have not encountered scenarios necessitating the use of random option generation.

The rationale behind not directly resorting to large language models for option extraction in the first stage stems from the superior stability and reliability of the rule-based method. Despite leveraging large language models for option extraction has been a common practice model evaluations, it still carries a certain error rate. On the contrary, the rule-based method, while not infallible in parsing answers across all scenarios, nearly guarantees correctness in the instances where parsing is successful. Consequently, we advocate for an initial screening using the rule-based method, followed by the employment of large language models for extraction, as a strategy that enhances overall robustness.

**Scoring**    In addition to the calculation of the GIA score mentioned above, our benchmark can also be broken down to calculate accuracy across various cognitive dimensions. Specifically, each question is annotated with the CHC factors it involves; factors that are involved are marked with a 1, and those that are not involved are marked with a 0. When a question involves a certain factor, the correctness of that question will contribute to the accuracy statistics for that particular CHC factor; otherwise, it will not be included in the statistics. Taking the calculation of the accuracy score of the Gc factor as an example:

$$Acc\_Gc = \frac{\sum_{i=1}^{n} Gc_i \cdot T_i}{\sum_{i=1}^{n} Gc_i} \tag{1}$$

where $n$ is the total number of questions, $Gc_i$ indicates whether the $i^{th}$ question involves the Gc factor, marked as 1 if it does, and 0 otherwise. $T_i$ indicates whether the $i^{th}$ question was answered correctly, with 1 representing a correct answer and 0 representing an incorrect answer. To mitigate the effects of randomness on the evaluation results, including both the scores of the various CHC factors and the overall GIA score, we adopt a strategy of iterating five times and taking the average.

## H  GIA SCORES ON MORE LANGUAGES

### H.1  ABLATION STUDY ON LLM SIZE

Table 5: **The training data and hyperparameters of MLLM with Qwen series.**

| Data and Hyperparameters | Pretrain | Finetune |
|---|---|---|
| data size | 558K | 1550K |
| batch size | 256 | 128 |
| lr | 1e-3 | 2e-5 |
| lr schedule | cosine decay | cosine decay |
| lr warmup ratio | 0.03 | 0.03 |
| weight decay | 0 | 0 |
| epoch | 1 | 1 |
| optimizer | AdamW | AdamW |

To further investigate the influence of LLM size to the GIA score, we conducted an ablation study with the Qwen series from 1.8B to 72B. In this experiment, we applied the LLaVA architecture and used the same ViT component (CLIP-ViT-L-14). In order to strictly control variables, we trained the models by ourselves using the same training data and the same set of hyperparameters for pretraining and fine-tuning. The data for pretraining is completely from LLaVA-1.5, and the data for fine-tuning is composed of LLaVA1.5 (Liu et al., 2023a) dataset, ShareGPT4v (Chen et al., 2023) dataset and our private visual-text instruct data. We show the training data and hyperparameters for both first-stage vision-language alignment pretraining and the second-stage visual instruction tuning in Table. 5. We use greedy decoding for evaluation to ensure reproducibility. The GIA scores on six languages are shown in Fig. 4.

Across the six languages analyzed, we consistently observe a significant increase in GIA scores with the expansion of LLM parameters. However, it is notably surprising that scaling up the size of LLMs

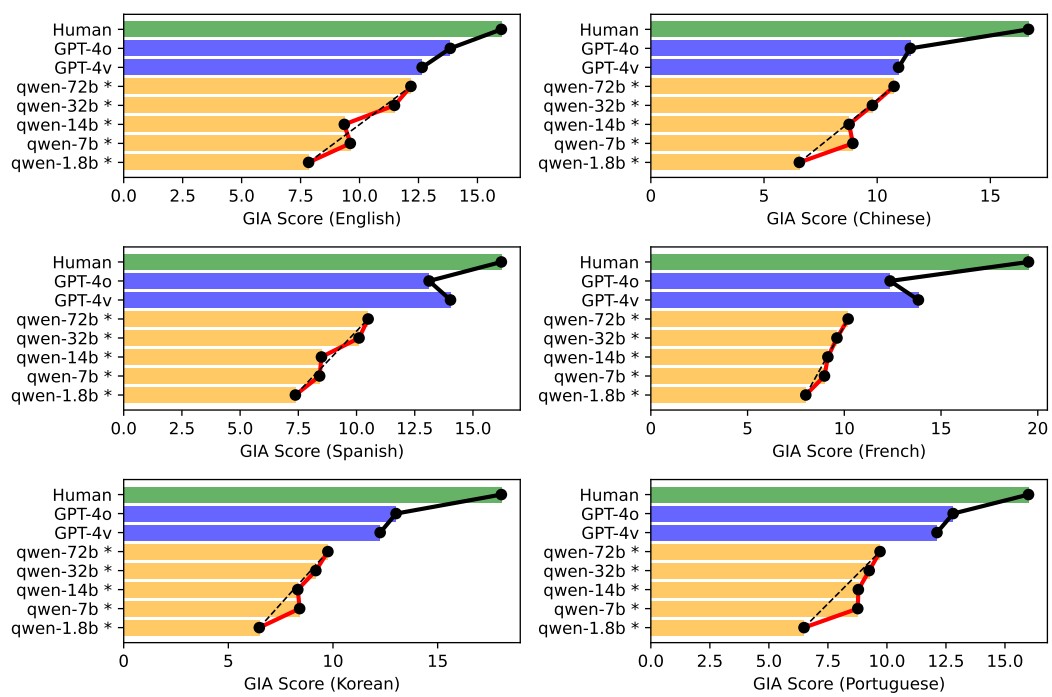

Figure 4: **The GIA scores across the six languages, with Qwen LLM series from 1.8B to 72B.** Generally, the GIA scores increase with the rise of LLM parameters. However, a threshold is observed when scaling up the LLMs' size from 7B to 14B.

from 7B to 14B parameters often yields no observable performance enhancement (and there might even be a slight decline). This phenomenon suggests the existence of a threshold - indicative of an emerging point of general intelligence for MLLMs somewhere between 13B and 32B parameters. In other words, it indicates a potential threshold for attaining a superior level of general intelligence, likely situated in the parameter range of 13B to 32B.

## H.2   GIA SCORE CAN BETTER REFLECT HUMAN PREFERENCE

We perform linear regression to calculate the $R^2$ correlation between models' scores on Chatbot Arena (Chiang et al., 2024) and their GIA scores from M3GIA. We also compared these correlations with scores obtained from traditional task-oriented benchmarks, such as MMMU (Yue et al., 2023), MMBench (Liu et al., 2023b), MM-Vet (Yu et al., 2023), and OCR-Bench (Liu et al., 2023c).

Table 6: **Models' *human preference score* on Chatbot Arena (Vision) and their scores on various benchmarks**, including MMMU, MMBench (MMB), OCRBench (OCRB) (Liu et al., 2023c), MMVet, and the average performance (Avg.)  across 8 prominent benchmarks: MMMU, MMB, HallusionBench (Guan et al., 2023), MMVet, OCRB, AI2D (Kembhavi et al., 2016), MMStar (Chen et al., 2024), MathVista. The GIA score of M3GIA is calculated as the average score across English and Chinese, as Chatbot Arena (Vision) only supports these two languages.

| Models | GPT-4o | Gemini-1.5-Pro | Claude-3-Sonnet | Claude-3-Haiku | GPT-4o-mini | Qwen2-VL-7B | MiniCPM-v2.6 |
|---|---|---|---|---|---|---|---|
| Arena Score | 1227 | 1220 | 1048 | 1000 | 1122 | 1053 | 975 |
| MMBench (Acc.) | 80.5 | 73.9 | 81.7 | 57.1 | 75.9 | 83.0 | 81.8 |
| MMMU (Acc.) | 69.2 | 60.6 | 66.4 | 49.7 | 60.0 | 54.1 | 49.8 |
| OCRBench (Acc.) | 80.5 | 75.4 | 64.6 | 65.8 | 78.5 | 84.5 | 85.2 |
| MM-Vet (Acc.) | 75.1 | 64.0 | 51.7 | 46.4 | 66.9 | 62.0 | 60.0 |
| Avg. Performance | 71.5 | 64.4 | 53.5 | 51.5 | 64.1 | 63.3 | 60.5 |
| M3GIA (GIA score)* | 92.4 | 78.1 | 72.5 | 71.2 | 75.6 | 74.3 | 65.6 |

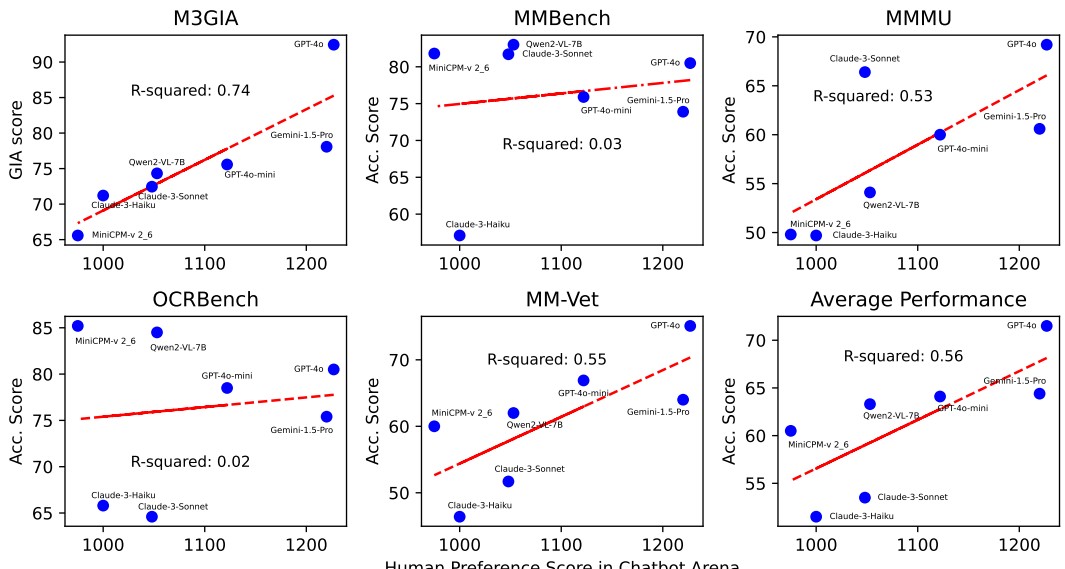

Figure 5: **Correlation between our GIA score and human preference (Chatbot Arena vision).** M3GIA aligns more effectively with actual human experience.

- **Strongest Correlation with Human Preference:** As shown in Fig. 5, our GIA score **indeed** demonstrated the strongest correlation with human preference scores on Chatbot Arena among all the benchmarks evaluated.

- **Benchmark Averaging as a Comparison:** In the current MLLM community, it is widely recognized that a single benchmark often fails to truly reflect model capabilities, leading to significant gaps between benchmark scores and actual human experiences. To address this, researchers commonly resort to averaging scores across multiple benchmarks, but this process is time-intensive and resource-heavy. To validate the significance of M3GIA, we calculated the average scores of the models across 8 prominent benchmarks, including MMMU (Yue et al., 2023), MMBench (Liu et al., 2023b), MM-Vet (Yu et al., 2023), OCR-Bench (Liu et al., 2023c), HallusionBench (Guan et al., 2023), AI2D (Kembhavi et al., 2016), MMStar (Chen et al., 2024), MathVista (Lu et al., 2023) and found: the average score across these benchmarks exhibited a higher correlation with human preference scores compared to individual benchmark scores, but the correlation between the average benchmark score and human preference ($R^2 = 0.56$) is still lower than the correlation between M3GIA's GIA score and human preference ($R^2 = 0.74$).

In summary, M3GIA achieves a level of correlation with human preferences. Crucially, it achieves this with just a single, unified test suite, significantly simplifying the evaluation process and addressing the pain point of benchmarking complexity in the MLLM community.

# I CASE STUDY

## I.1 THE COMMON SENSE CLUSTER

Current advanced MLLMs excel in common sense cluster, especially in general information and oral vocabulary questions, which can likely be bolstered by their extensive training datasets. However, there are still some deficiencies in logo problem related to cultural background for some MLLMs, e.g. GPT-4v. Logo problems usually contain confusing artistic characters or symbolic expression of cultural elements, which requires a high level of Gc and a certain level of Gv. As shown in Fig. 6, GPT-4v can recognize the locomotive in the logo of chinese question 46, but it fails to recognize the Chinese character ("hang" in pinyin) in chinese question 41, while GPT-4o can perfectly recognize characters containing Chinese cultural elements.

## Common Sense Cluster

### Oral Vocabulary | Gc | Pt.7

**Question**: Nas questões a seguir escolha a palavra que melhor expressa o significado da palavra dada. (Portuguese)

ACORDADO
A. Iluminado
B. Percebido
C. Abalado
D. Despertado

**Choices**:
(A) Iluminado
(B) Percebido
(C) Abalado
(D) Despertado
**GT**: D

GPT-4O: (D) Despertado ✓
GPT-4V: (D) Despertado ✓

### General Information | Gc | En.35

**Question**: Where can you find the building featured on this note?

**Choices**:
(A) Washington DC
(B) London
(C) Philadelphia
(D) Atlanta
**GT**: C

GPT-4O: (C) Philadelphia ✓
GPT-4V: (C) Philadelphia ✓

### Logo Problem | Gc, Gv | Zh.41

**Question**: 这是中国西南部一座城市的形象标志，这个标志代表的汉字是哪个?

**Choices**:
(A) 沪
(B) 汴
(C) 榕
(D) 杭
**GT**: D

**GPT-4O**: (D) 杭 ✓
Explanation: The image shows a stylized representation of a Chinese character particularly in a modern or artistic form. The character in the image appears to be "杭," which is the abbreviation for the city of Hangzhou (杭州) in China.

**GPT-4V**: (A) 沪 ✗

### Logo Problem | Gc, Gv | Zh.46

**Question**: 图为中国一个国有集团的标志，请问与该标志最为形似的事物是哪项?

**Choices**:
(A) 火车头 (locomotive)
(B) 盾牌 (Shield)
(C) 太阳 (Sun)
(D) 轮子(Wheel)
**GT**: A

**GPT-4O**: (A) 火车头 (locomotive) ✓
Explanation: The image represents the China Railway logo, which prominently features the front of a locomotive or train head at its core. This indicates that choice (A) is the most appropriate one for what the symbol most closely resembles.

**GPT-4V**: (A)火车头 ✓

Figure 6: **Case Study of Common Sense Cluster.**

| Visual-spatial Cluster |
| --- |

| Visualization \| Gv \| En.80 |
| --- |

**Question**: In the Block Rotation subtest, you will see five blocks of different 3D shapes. The block on the top shows the target block. One of the four blocks labeled A through D also shows the target block, but rotated in space. The remaining blocks show a different block, rotated to various positions in space. Your task is to choose the block (from the blocks labeled A through D) that shows the target block rotated in space. Please select the correct option.

**Choices**:
(A)   1
(B)   2
(C)   3
(D)   4

**GT**: D

GPT-4O: (C) 3   ✗
Explanation:
The task is to identify which block (from A to D) is a rotated view of the target block shown at the top.
Here's the detailed analysis:
Target Block Description:
    Top block: Pink
    Middle layer:
    Left block: Blue
    Right block: Yellow
Bottom layer:
    Left block: Blue
    Right block: Green
Options Analysis:
    Option 1 (A) shows a horizontal layout, not matching the vertical arrangement of the target block.
    Option 2 (B) has a vertical layout but the arrangement of colors and blocks does not match the target.
    Option 3 (C) correctly shows a vertical arrangement where the:
        The top block is pink.
        The middle layer on the left is blue and the right is green.
        The bottom layer on the left is pink, the right is blue.
    Option 4 (D) also shows a vertical arrangement but the relative positions of colors do not match exactly.
Conclusion:
After examining the structure and rotational perspectives, Option 3 (C) is the correct choice that represents the target block rotated in space.
Thus, the correct answer is:
Choices: (A) 1 (B) 2 (C) 3 (D) 4
Answer: (C) 3
Option (C) correctly maintains the spatial arrangement of the target block when rotated. The top is pink, middle includes blue and green, and the bottom has pink

Figure 7: **Case Study of Visualization problem in Visual-Spatial Cluster.**

I.2   THE VISUAL-SPATIAL CLUSTER

In the Visual-spatial Cluster, current advanced MLLMs performe very well on the Picture Recognition questions, followed by the Real-world Spatial questions, and performed the worst on the Visualization

transformation questions. The high accuracy on the Picture Recognition questions shows that the advanced MLLMs already has a good object recognition ability. Compared with object recognition ability, their ability to recognize three-dimensional spatial relationships is much worse, which can be divided into translation transformation and rotation transformation. The performance on the Real-world Spatial questions proves that the MLLMs can recognize the translation transformation relationship of objects in three-dimensional space with a certain probability, including up, down, left, right, front, and back. At the same time, the MLLMs suffer from the rotation transformation ability and spatial imagination ability in three-dimensional space, resulting the low accuracy on the Visualization transformation questions. As shown in Fig. 8, after multiple inferences, GPT-4o can always recognize the same cup in english question 81 and the spatial relationship between the two remote controls with a high probability in english question 99, but it is difficult to recognize the same blocks after rotation in english question 80 in Fig. 7.

### I.3   THE COMPREHENSION CLUSTER

Similar to the common sense cluster, current advanced MLLMs perform very well in comprehension cluster, including readings-text, readings-VL and the comic problem, which can be attributed to the powerful language capabilities of LLM. Surprisingly, GPT-4o understands the scenarios represented both visually and textually in comics quite well, which proves it can integrate visual narrative comprehension with textual comprehension. As shown in Fig. 10, in english question 146 and french question 144, GPT-4o can understand the entire story's connotation based on the textual dialogues between characters and the plot development, especially can recognize the facial expressions and quantitative contrast of population in english question 146. At the same time, GPT-4o still has some shortcomings in understanding the relationship between text paragraphs. As shown in Fig. 9, in english question 7, GPT-4o fails to capture the "general-specific-general" structure of the article.

### I.4   THE MATHEMATICS CLUSTER

This Mathematics cluster is designed to evaluate the Gq factor. Although current advanced MLLMs did not perform well on math problems overall, we found two interesting phenomena. One is that the model performs better on algebra problems than geometry problems, such as the english question 182 in Fig. 11. This may be attributed to the training data of LLM contains enough math knowledge text, but the visual module of MLLMs still has defects in abstract geometric figures and their relationships. The other is that the model performed better on math facts problems and problems that can be solved in one step by directly applying mathematical knowledge including symbolic knowledge and geometric knowledge than on problems that require multi-step reasoning. For example in Fig. 11, GPT-4o can apply the Central Angle Theorem to solve the english question 182, but fails to solve the english question 175 which needs multi-step reasoning and calculation. In addition, GPT-4o has reached a level of practical application in simple mathematical applied problem, such as the problem of choosing the shortest flight time in english question 188 as shown in Fig. 12.

### I.5   THE REASONING CLUSTER

The reasoning cluster is designed to evaluate the I (inductive) factor and RG (deductive reasoning) factor. Similar to the performance gap between geometry and algebra in mathematics cluster, there is also a performance gap between deductive and inductive reasoning. Although GPT-4o are approaching the average human level for deductive reasoning, it only marginally meet the passing line (60) on syllogism problem and real-world reasoning problem. For example in Fig. 14, GPT-4o fails on the english question 286 which is a classic form of deductive reasoning and ask GPT-4o to decide which of the given conclusions logically follows from the two given statements. For inductive reasoning, GPT-4o performs quiet well in number series and concept formation problems, such as the english question 214 and 237 in Fig. 13, but performs very poorly on the Raven's Matrices problems. Take the english question 254 in Fig. 15 as an example, GPT-4o mistakenly recognized the graphic in the third row and first column as a vertical line with a black square at the bottom, when it should actually be a black square at the top, resulting in the incorrect selection. GPT-4o can perform effective reasoning, but there is a certain probability that it will make small mistakes when recognizing graphics, which shows that its visual module needs to be further improved. In addition, we also show the results of GPT-4v, which misidentifies counterclockwise rotation as clockwise rotation and incorrectly identifies

option E as the arrow pointing straight down. This proves GPT-4v is much worse than GPT-4o in both reasoning and visual recognition.

## J  DISCUSSION

The strong correlations among MLLMs' accuracy scores across cognitive dimensions suggest the presence of a low-dimensional latent variable, akin to the human g factor, that explains variance in performance (Murtazina & Avdeenko, 2021). This supports the notion that MLLMs exhibit a general cognitive ability influencing their task performance. Furthermore, GIA scores, derived using a CFA model based on human data, show high predictive power for overall model accuracy ($R^2 \geq 0.93$), confirming the relevance of the g factor in assessing AI performance across languages (Burkart et al., 2017; Dubois et al., 2018; McGrew, 2009). These findings indicate that the g factor is a meaningful metric for evaluating AI "intelligence," providing a unified framework for comparing and improving models. The successful application of human cognitive models to MLLMs suggests valuable cross-domain insights, advancing both AI evaluation and our understanding of cognitive processes in artificial systems. We have observed a phenomenon in MLLMs similar to human cognition known as "winner takes all", which corroborates the emergence of GIA within cutting-edge MLLMs. However, we have not yet been able to provide a more definitive and persuasive explanation for the underlying causes. Unraveling this will be one of the directions we dedicate ourselves to in the future.

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

**Visual-spatial Cluster**

**Picture Recognition | Gv | En.81**

**Question**: There is a target object on the horizontal line. Find two objects under the horizontal line that are consistent with the target object.
.

**Choices**:
(A)   1, 2
(B)   2, 5
(C)   3, 4
(D)   4, 5

**GT**: B

GPT-4O: (B) 2, 5 ✓
Explanation**:** To find the objects consistent with the target object, we need to compare their characteristics such as shape, color, and logo:
•The target cup is white with a certain design and a Starbucks logo.
•Object 1: White cup, same logo.
•Object 2: White cup, same logo, same shape.
•Object 3: Cream cup, same logo but different shape.
•Object 4: White cup, different logo ("Dumb Starbucks").
•Object 5: White cup, same logo, same shape.
Therefore, objects 2 and 5 are consistent with the target object in terms of both logo and shape. Option (B) 2, 5 is the correct answer.

**Real-world Spatial | Gv | En.99**

**Question**: What position is the black remote control located in relation to the white remote control?

**Choices**:
(A)   front side
(B)   back side
(C)   Above
(D)   below

**GT**: B

GPT-4O: (C) above ✗
Explanation: From the provided image, the black remote control is situated above the white remote control.
Explanation: When observing the image, the black remote control is positioned on the armrest of the sofa above the white remote control.

GPT-4O: (B) back side ✓
Explanation: In the image, the black remote control is positioned closer to the upper portion of the image and behind the white remote control. This indicates that the black remote control is on the back side in relation to the white remote control.

Figure 8: **Case Study of Picture Recognition and Real-world Spatial problem in Visual-Spatial Cluster.**

## Comprehension Cluster

### Readings-VL | Grw | En.127

**Question**:
Passage: P1: Some people learn a second language easily. Others have trouble learning a new language. How can you learn a new language, such as English? There are several ways to help you learn English more easily. P2: Firstly, feel positive about learning English. If you believe that you can learn, you will learn. Be patient. You don't have to understand everything all at once. Often you will make mistakes when you are learning something new. We can learn from our mistakes. P3: Try to practice using your English as possible as you can. For example, you can write a diary every day. Soon, you will get used to writing your ideas in English. After several weeks, your writing skills will improve. Besides, try to speak English every day. You can practice speaking with your classmates after class. You might make mistakes, but don't worry. Slowly, you will become comfortable communicating in English. P4: It's a great idea to keep a record of your language learning experience. You can write your learning experience in your diary. After each class, think about it. Do you answer questions correctly in class? Do you understand your teacher? Perhaps the lesson is a bit difficult, but you can try to understand it. Write these reflections in your diary to practice using your English and write down your little progress. Finally, you will find yourself enjoy learning English. P5: All in all, be positive, confident and patient -- believe that you can make it sooner or later. Make good use of all the time we can get.
Question: Which image best describes the structure of this passage?
**Choices**:
(A)   A
(B)   B
(C)   C
(D)   D
**GT**: A

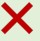

GPT-4O: (D) D
The passage provided describes a clear, logical progression of steps for learning English, presented in a sequential manner. Let's break down the paragraphs:
> P1: Introduces the topic of learning a new language and sets up the context.
> P2: Discusses the importance of feeling positive about learning English.
> P3: Emphasizes practicing English in various ways to improve gradually.
> P4: Suggests keeping a record of the language learning experience.
> P5: Concludes with a summary, encouraging positivity, confidence, and patience.
> The structure of the passage is linear, with each paragraph building upon the previous ones in a step-by-step manner.

Considering the images:
> Image A shows a more interconnected structure that isn't purely linear.
> Image B shows a similar structure but still has some intersections.
> Image C shows a purely parallel structure with equal importance given to P2 through P5, all stemming directly from P1, which is not quite accurate as each subsequent paragraph builds on the idea of the previous one.
> Image D shows a strictly sequential structure where each paragraph follows logically from the previous one.

Given the sequential nature of the passage:
The best choice is (D) D, as it accurately represents the step-by-step structure of the passage.

Figure 9: **Case Study of Readings-VL in Comprehension Cluster.**

## Comprehension Cluster

### Comic Problem | Grw, Gv | En.146

**Question**: This is a single choice question. Please choose the option that best fits the description of this comic?
**Choices**:
(A)   People in different countries define city size differently.
(B)   The boy yearns for the girl's hometown.
(C)   The boy was shocked by the small population of the girl's hometown.
(D)   Southern China is very prosperous
**GT**: C

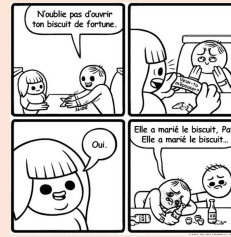

GPT-4O: (C) The boy was shocked by the small population of the girl's hometown. ✓
Explanation: In the comic, the boy asks the girl which part of China she comes from and she responds that she comes from a small city in the south. When the boy learns that the population is around five million, he is visibly shocked. This indicates that the boy finds five million to be a large number for a "small city," highlighting the difference in perception regarding city size. Therefore, option (C) is the most fitting description.

### Comic Problem | Grw, Gv | Fr.144

**Question**: Pourquoi le garçon était-il triste à la fin ?
Translation: Why is the boy sad at the end? (French)
**Choices**:
(A)   Parce que la jeune fille n'était pas d'accord avec le contenu de la note.
(B)   Parce que la fille a cassé son cookie.
(C)   Parce que la fille a pris la note au pied de la lettre et a épousé le cookie.
(D)   Parce que la fille est tombée amoureuse d'autres hommes.
**GT**: C

GPT-4O: (C) Parce que la fille a pris la note au pied de la lettre et a épousé le cookie. ✓
Explanation:
Option A: This suggests that the girl disagreed with the content of the note. However, the comic doesn't depict any disagreement; rather, it shows confusion and literal interpretation.
Option B: This suggests that the girl broke the cookie. This is not depicted in the comic; instead, the focus is on the literal interpretation of the note in the cookie.
Option C: This suggests that the girl interpreted the note literally and decided to marry the cookie. This is in line with the comic, where the girl says "Oui" to the note's proposal and the boy is upset, lamenting that she married the cookie.
Option D: This suggests that the girl fell in love with other men. There's no such detail in the comic.
The best answer based on the interpretation of the comic:
Answer: C. Parce que la fille a pris la note au pied de la lettre et a épousé le cookie.
This answer fits the context where the girl misunderstood the proposal in the fortune cookie and decided to take it literally, leading to the boy's sadness.

Figure 10: **Case Study of Comic Problem in Comprehension Cluster.**

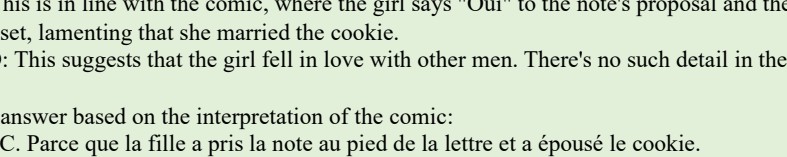

## Mathematics Cluster

### Math Facts | Gq | En.182

**Question**: If angle AOB is 40 degrees, what is the measure of angle ACB?
**Choices**:
(A)  10
(B)  20
(C)  30
(D)  40
**GT**: B

GPT-4O: (B) 20 ✓
Explanation: In a circle, when an angle is formed at the center (angle AOB) and another angle is formed on the circumference (angle ACB) by the same arc, the angle at the center is twice the angle at the circumference. This is known as the Central Angle Theorem.
Given: Angle AOB = 40 degrees
According to the Central Angle Theorem: Angle AOB = 2 * Angle ACB
Therefore, 40 degrees = 2 * Angle ACB Angle ACB = 40 degrees / 2 Angle ACB = 20 degrees
So, the measure of angle ACB is 20 degrees..

### Geometry | Gf, Gv, Gq, RQ | En.175

**Question**: In the figure below, lines d and f are parallel and the angle measures are as given. What is the value of x?
**Choices**:
(A)  55
(B)  75
(C)  85
(D)  90.
**GT**: C

GPT-4O: (A) 55 ✗
Explanation:Given that lines ( d ) and ( f ) are parallel and the angle measures are as given, we need to find the value of ( x ).
Since ( d \parallel f ), the corresponding angles formed by the transversal ( h ) are congruent.
The 120° angle and the angle adjacent to ( x ) (let's call it angle ( y )) are corresponding angles.
Let's break it down step by step:
Find the value of angle ( y ): Since the 120° angle is a corresponding angle to ( y ): [ y = 120° ]
Observe that angle ( y ) and ( x ) form a linear pair (they are on a straight line): [ x + y = 180° ]
Substituting ( y = 120° ): [ x + 120° = 180° ] [ x = 180° - 120° ] [ x = 60° ]
The value of ( x ) is 60°. However, there is an inconsistency here as ( 60° ) is not an option among the given choices, and we see that 35° and ( x ) should add up to make 90° due to the perpendicular formed at the intersection.
Thus, looking at the diagram closely again, consider whether there might be an error with the provided options or diagram interpretation. Given how we used the provided values to solve through standard geometric relationships, it's plausible to question the integrity or context of the problem statement. Note that: [ x = 35° + y ] where it seems (y = 55°) should make angles: [ x + y (35°) = 90° as a vertical angle adjustment correction. ]
In intuitive term, actual solving could lean:
The correct value should logically be 55 option A.

Figure 11: **Case Study of Readings-VL in Mathematics Cluster.**

## Mathematics Cluster

### Applied Problem | Gq, RQ | En.188

**Question**: The picture is a math-related question. Please answer this question and choose the best option?

**Choices**:
(A)   Flight 1
(B)   Flight 2
(C)   Flight 3
(D)   They take exactly the same time

**GT**: C

| | | |
|---|---|---|
| 11:45am - 2:28pm
New York (JFK) - Las Vegas (LAS)
▲ Delta
Carry-on included | **Flight 1** | $267
Roundtrip per traveler |
| 7:30am - 10:06am
New York (JFK) - Las Vegas (LAS)
▲ Delta
Carry-on included | **Flight 2** | $267
Roundtrip per traveler |
| 10:30am - 1:02pm
New York (JFK) - Las Vegas (LAS)
✈ JetBlue Airways | **Flight 3** | $270
Roundtrip per traveler |

**GPT-4O: (C) Flight 3** ✓

To determine which flight takes the shortest time, we need to calculate the duration of each flight.

•**Flight 1:** 11:45 am - 2:28 pm
   •   Duration: 2:28 pm - 11:45 am = 2 hours and 43 minutes
•**Flight 2:** 7:30 am - 10:06 am
   •   Duration: 10:06 am - 7:30 am = 2 hours and 36 minutes
•**Flight 3:** 10:30 am - 1:02 pm
   •   Duration: 1:02 pm - 10:30 am = 2 hours and 32 minutes

Comparing these durations:
•Flight 1: 2 hours and 43 minutes
•Flight 2: 2 hours and 36 minutes
•Flight 3: 2 hours and 32 minutes

The shortest duration is for Flight 3.

### Algebra | Gq, RQ | En.156

**Question**: The picture is a math-related question. Please answer this question and choose the best option?

14. If $f(x) = x$ and $g(x) = \sqrt{x}, \quad x \geq 0$, what are the solutions of $f(x) = g(x)$

**Choices**:
(A) x = 1  (B) x1 = 1, x2 = -1  (C) x1 = 1, x2 = 0  (D) x = 0  (E) x = -1

**GT**: C

**GPT-4O: (C) x1 = 1, x2 = 0** ✓

To solve the given problem, we need to find the values of ( x ) that satisfy ( f(x) = g(x) ):
Given: [ f(x) = x ] [ g(x) = \sqrt{x}, \quad x \geq 0 ]
We need to solve ( f(x) = g(x) ): [ x = \sqrt{x} ]
Square both sides of the equation to eliminate the square root: [ x^2 = x ]
Rearrange the equation to solve for ( x ): [ x^2 - x = 0 ] [ x(x - 1) = 0 ]
Set each factor equal to zero: [ x = 0 ] [ x - 1 = 0 \rightarrow x = 1 ]
Therefore, the solutions are ( x = 0 ) and ( x = 1 ).
Given the choices: (A) x = 1 (B) x1 = 1, x2 = -1 (C) x1 = 1, x2 = 0 (D) x = 0 (E) x = -1
The correct choice is:
(C) x1 = 1, x2 = 0
Explanation: The roots of the equation ( x^2 - x = 0 ) are ( x = 0 ) and ( x = 1 ), which correspond to the choices listed as ( x1 = 1, x2 = 0 ). Hence, option (C) is correct.

Figure 12: **Case Study of Comic Problem in Mathematics Cluster.**

## Reasoning Cluster

### Concept Formation | I | En.237

**Question**: Choose a different type of graph than the other graphs?
(A)   1
(B)   2
(C)   3
(D)   4
(E)   5
**GT**: D

GPT-4O: (D) 4 ✓
Explanation: The given task is to identify which item among the presented choices is of a different type compared to the others.
From the image:
•Choices 1, 2, 3, and 5 all show wristwatches.
•Choice 4 shows a wall clock.
Therefore, the graph that is of a different type is the wall clock.

### Number Series | I, RQ | En.214

**Question**: Look at this series: 22, 21, 23, 22, 24, 23, ... What number should come next?
**Hint:** In each series, look for the degree and direction of change between the numbers. In other words, do the numbers increase or decrease, and by how much?
**Choices**:
(A)   22
(B)   24
(C)   25
(D)   26
**GT**: C

Look at this series: 22, 21, 23, 22, 24, 23, ... What number should come next?
Ⓐ 22
Ⓑ 24
Ⓒ 25
Ⓓ 26

GPT-4O: (C) 25 ✓
Explanation: The given series is: 22, 21, 23, 22, 24, 23, ...
To find the next number in the series, we should look for a pattern in the changes between the numbers. Let's examine the pattern step by step:
•The first number is 22.
•The pattern of changes: -1, +2, -1, +2, -1
Following this pattern:
•22 (initial value)
•21 (22 - 1)
•23 (21 + 2)
•22 (23 - 1)
•24 (22 + 2)
•23 (24 - 1)
Given that the pattern between the numbers involves alternating between subtracting 1 and adding 2, the next number should follow this pattern:
•The last number is 23.
•According to the pattern, after subtracting 1 and adding 2 alternately, the next step after subtracting 1 (which resulted in 23) should be to add 2.
Therefore, 23 + 2 = 25.

Figure 13: **Case Study of Number Series and Concept Formation problem in Reasoning Cluster.**

## Reasoning Cluster

### Syllogism Problem | RG | En.286

**Question:** In each question below are given two statements followed by two conclusions numbered I and II.
You have to take the given two statements to be true even if they seem to be at variance from commonly known facts.
Read the conclusion and then decide which of the given conclusions logically follows from the two given statements, disregarding commonly known facts.
Statements:
      All good athletes win.
      All good athletes eat well.
Conclusions:
      I. All those who eat well are good athletes.
      II. All those who win eat well.

**Choices**:
(A)   Only conclusion I follows
(B)   Only conclusion II follows
(C)   Either I or II follows
(D)   Neither I nor II follows
(E)   Both I and II follow

**GT**: D

> Statements: All good athletes win. All good athletes eat well.
> Conclusions:
> I. All those who eat well are good athletes.
> II. All those who win eat well.
>
> A. Only conclusion I follows
> B. Only conclusion II follows
> C. Either I or II follows
> D. Neither I nor II follows
> E. Both I and II follow

GPT-4O: (B) Only conclusion II follows    ✗
Explanation: Let's analyze the given statements and conclusions:
Statements:
All good athletes win.
All good athletes eat well.
Conclusions:
I. All those who eat well are good athletes. II. All those who win eat well.
Analysis:
From the first statement "All good athletes win," we know that if someone is a good athlete, they will win.
From the second statement "All good athletes eat well," we know that if someone is a good athlete, they will eat well.
Conclusion I: "All those who eat well are good athletes."
This does not logically follow from the statements. The statements only tell us that good athletes eat well, not that everyone who eats well is a good athlete. Therefore, conclusion I does not follow.
Conclusion II: "All those who win eat well."
This can be derived from the given statements. If all good athletes win and all good athletes eat well, then all those who win (presumably good athletes) would also eat well. Therefore, conclusion II follows.
Thus, the correct choice is:
Choices: (A) Only conclusion I follows (B) Only conclusion II follows (C) Either I or II follows (D) Neither I nor II follows (E) Both I and II follow
Answer: (B) Only conclusion II follows

Figure 14: **Case Study of Syllogism Problem in Reasoning Cluster.**

Reasoning Cluster

Raven's Matrices | I, Gv | En.257

**Question:** In the question, you'll see a 3x3 matrix of geometric shapes with one piece missing. Please select the correct tile from a collection of six options (labeled through A to F) to complete the general pattern in the matrix by replacing the question mark with the geometric shape you select. Which is the correct option?
**Choices**:
(A) A. (B) B  (C) C  (D) D  (E) E. (F) F
**GT**: F

GPT-4O: (F) F ✓
Explanation: To solve this puzzle, we need to identify the pattern each row follows and then see how the missing piece fits.
Analysis:
    First row (Horizontal shapes):
        First shape: Square on the right
        Second shape: Square in the middle
        Third shape: Square on the left
    Second row (Diagonal shapes):
        First shape: Square at one end (tilted down-left)
        Second shape: Square in the middle (tilted)
        Third shape: Square at one end (tilted down-right)
    Third row (Vertical shapes):
        First shape: Square at the bottom
        Second shape: Square in the middle
        Third shape: Missing
Pattern:
The pattern within each row transitions the position of the square end, middle, opposite end.
Conclusion:
For the third row (vertical shapes), the square placement should be at the top.
Reviewing the Options:
A: Diagonal bar, square end on top-left corner.  B: Vertical bar, square in the middle
C: Horizontal bar, square on the right.   D: Horizontal bar, square in the middle
E: Diagonal bar, square end on bottom-right.  F: Vertical bar, square on top
Correct Answer: (F)

GPT-4V: (E) E ✗
Explanation: The 3x3 matrix displays patterns that change by column and row. To solve puzzles like this, you usually have to look for patterns or rules that apply across the rows and columns. Here's how the patterns work in this matrix:
First column: The shapes are rotated 45 degrees clockwise with each subsequent cell.
Second column: The shapes also follow the 45-degree clockwise rotation rule.
Third column: The first cell in this column is rotated 45 degrees clockwise, so we can deduce that the shapes in this column should follow the same pattern.
By applying this logic, the missing shape in the last cell should be the arrow pointing straight down, after being rotated 45 degrees clockwise from the previous shape. Out of the given options, (E) is the arrow pointing straight down, fitting into this pattern correctly. Hence, option (E) is the correct answer.

Figure 15: **Case Study of Raven's Matrices Problem in Reasoning Cluster.**