# OpenReview forum: "M3GIA: A Cognition Inspired Multilingual and Multimodal General Intelligence Ability Benchmark"
_ICLR.cc/2025/Conference — ICLR 2025 Conference Withdrawn Submission_

### Official Review · Reviewer_1Zrf · 2024-10-19

**Soundness:** 3
**Presentation:** 2
**Contribution:** 2
**Rating:** 5
**Confidence:** 4

**Summary:**

This paper introduces the concept of a cognitive-driven, multilingual, and multimodal benchmark to evaluate the general intelligence of MLLMs, referred to as M3GIA. The benchmark is grounded in the well-established Cattell-Horn-Carroll (CHC) model of intelligence and proposes a novel evaluation metric. It is open-sourced and aims to enhance the cognitive capabilities of MLLMs.

**Strengths:**

1. The use of a taxonomy of cognitive abilities to evaluate the general intelligence of MLLMs is good, as it enables a more systematic evaluation.
2. The benchmark is constructed using unpublished offline data, which is a good practice to prevent data leakage.
3. The benchmark includes multiple language variants, allowing for the evaluation of MLLMs’ general intelligence across different languages.

**Weaknesses:**

1. While the paper mentions that several specific factors from the CHC model of intelligence were selected (lines 237-250), it is unclear why these particular factors were chosen and how they relate to the general intelligence of MLLMs.
2. Although incorporating cognitive science into the evaluation of MLLMs is a positive step, the underlying tasks remain traditional, such as Math, Logo Problem, and Comic Problem. This may detract from the benchmark’s novelty. Given that recent works like MMMLU also include multilingual variants [1], it is not clear how M3GIA is fundamentally different from MMMLU.
3. The paper introduces numerous cognitive concepts and abbreviations, which may make it difficult for readers unfamiliar with cognitive science to follow. For instance, the meaning of “Fluid Reasoning (Gf)” (line 97) in the context of MLLMs is not clearly explained. In my personal aspect, I feel odd about the term "Fluid Reasoning (Gf)" what does it mean?

[1] https://huggingface.co/datasets/openai/MMMLU (Multi-Language Variant of MMMLU)

**Questions:**

1. Is it possible to reorganize existing multi-task, multimodal benchmarks (e.g., MMMLU) to follow the taxonomy of cognitive abilities to evaluate the general intelligence of MLLMs? If not, could you explain why? Does the MMMLU benchmark lack specific tasks that would prevent it from capturing certain cognitive abilities?

[1] https://huggingface.co/datasets/openai/MMMLU (Multi-Language Variant of MMMLU)

---

> ### Author Response · Authors · 2024-11-21
>
> Thank you for your time and effort in reviewing our paper. It is encoraging to hear that you appreciated our practice of using unpublished offline data to prevent data leakage and recognized the use of the CHC taxonomy as meaningful. We address your concerns below.
>
> > **Concern1. Why these particular factors were chosen and how they relate to the general intelligence of MLLMs.**
>
> In fact, we have a very detailed discussion on this question in Supplementary Material.
>
> Please refer to **Appendix. A3 "HOW THE FIVE FACTORS ARE CHOSEN FOR EVALUATING MLLMS?"** for more details.
>
> > **Concern2. How M3GIA is fundamentally different from MMMLU**
>
> Firstly, we wish to confirm whether the MMMLU you mentioned refers to OpenAI's work:
> > Multilingual Massive Multitask Language Understanding (MMMLU) (https://huggingface.co/datasets/openai/MMMLU)
>
> If so, this benchmark is text-only and not a multimodal benchmark. The 'MM' in its name refers to 'Massive Multitask,' not 'Multimodal.'
>
> To the best of our knowledge, as of the time of our submission, M3GIA and M3Exam are the only multilingual and multimodal benchmarks.
>
> > **Concern3. Difficult for readers unfamiliar with cognitive science to follow.**
>
> Thank you for your feedback. We fully understand that colleagues without a psychology background might find some CHC-related concepts confusing, especially regarding the specific definitions of the factors. Therefore, **we have specifically dedicated an entire section in the appendix to introducing the definitions of these CHC concepts. Please refer to Appendix B: DEFINITIONS OF THE CHC FACTORS.**
>
> As these concepts require rigorous and clear explanations, the content is quite lengthy, making it difficult to include in the main text. Thus, we chose to place it in the appendix.
>
> To make our content clearer, we added a reference in the main text at line 213: “For detailed introductions and specific definitions of CHC factors, please refer to Appendix B.”

---

> > ### Comment · Reviewer_1Zrf · 2024-11-21
> >
> > Did I miss anything? At now, it seems your submission does not have an appendix with only page 14

---

> > ### Comment · Reviewer_1Zrf · 2024-11-22
> >
> > Sorry, I made an incorrect reference. What I meant to refer to is MMMU (https://arxiv.org/pdf/2311.16502).
> >
> > I understand that compared to MMMU, M3GIA is multilingual, which is good.
> >
> > However, I still want to know: aside from the multilingual setting, how does your benchmark differ from MMMU?
> > Is it possible to **reorganize existing multi-task, multimodal benchmarks (e.g., MMMU) to follow the taxonomy of cognitive abilities** in order to evaluate the general intelligence of MLLMs?
> >
> > As it seems in M3GIA, the underlying tasks remain traditional, such as Math, Logo Problems, and Comic Problems based on Figure 3 of your submission.

---

> ### Author Response · Authors · 2024-11-21
>
> Oops! The supplementary material for ICLR this year is not directly appended to the main text but **is provided as a separate document.**
>
> Please **click the PDF link under 'Supplementary Material'** to view the appendix (located below the 'Abstract' and above the 'Primary Area' section).

---

> ### Author Response · Authors · 2024-11-22
>
> Thanks for your reply. We wish to clarify that our question types are actually very different to MMMU or any other existing multi-task, multi-modal benchmarks.
>
> Most of our question types, such as Concept Formation, Visualization, Picture Recognition, and Syllogism Problems, are specifically tailored to assess the relevant CHC factors.
>
> **These question types have never appeared in other multi-modal benchmarks before, making it unfeasible to reorganize existing benchmarks to follow the CHC taxonomy for evaluating the general intelligence of MLLMs.** (Please refer to the Table2, 3 in the Appendix for the detailed description on these question types.)
>
> Some of our questions may happen to share similar names with those in other benchmarks, which might have led to the misunderstanding that we are using the same types of questions. However, they are only similar in name, and the content is entirely different. For example, 'Visual Reasoning' in SEEDBench seems similar to our Visual cluster. But in fact, their questions involve providing a photo of a daily life scenario and asking the model to infer what is happening. In contrast, in our Visualization task, the model is asked to identify the rotated 3D block that matches the original 3D block or is required to identify pieces that form a complete target shape.
>
> Of course, while most of M3GIA's question types are novel, there are indeed some questions with similar types found in other benchmarks. However, these are limited to General Info[1][2], Math[3][4], and Logo Problems. *(We disagree that comic problem is a traditional and widely known task in the LLM community. We have checked the current mainstream MLLM benchmarks, including MME, MMVET, MMMU, M3Exam, MMBench, TextVQA, Seedbench, DocVQA, OCRBench, RealworldVQA, ChartQA, and BigBench, and did not find tasks involving understanding multi-panel comics rich in text. The only comic-related work we acknowledge is CoMix[5], which is released after our submission.)*
>
> The reason for selecting them is that they align very well with the definitions of the respective CHC factors (General Info and Logo for Gc, Math for Gq). For these 3 types of questions, **if we set aside the issue of multilingual support and consider only English,** it is indeed possible to integrate some questions from other benchmarks as material for these types of questions.
>
> However, we believe that providing the community with more new, high-quality data is itself a valuable contribution. Moreover, the questions in the other five languages cannot be obtained from any existing benchmark.
>
> **We would like to reiterate that, to the best of our knowledge, no benchmark simultaneously includes all of our question types in a way that could be reorganized into our test.**
>
> ## The difference with MMMU
> MMMU is a human disciplinary test **that heavily relies on domain-specific knowledge** across six disciplines, including Art, Business, Health & Medicine, Science, Humanities & Social Science, and Tech & Engineering. **Its measurement results largely depend on the model's domain knowledge rather than intelligence itself.**
>
> Just as school subject exams are not comparable to IQ tests in evaluating students' intelligence, subject exams are greatly influenced by education level rather than purely by IQ. For instance, a child growing up in a poor region may possess high IQ but might perform worse on subject tests compared to a less intelligent but well-educated student due to a lack of educational resources.
>
> In contrast, M3GIA, except for the Gq cluster (Math), consists of cognitive test questions designed to minimize reliance on prior knowledge and focus solely on intelligence. For example, in our Raven's Matrices questions, the model is only provided with abstract patterns and required to identify the rules to fill in the blanks. **This is completely decoupled from domain knowledge, offering a more authentic reflection of the model's 'IQ' factor.**
>
> The reason we use math problems to assess Gq is that Gq is a relatively unique factor. Its definition in CHC is: The depth and breadth of knowledge about mathematics and the ability to comprehend quantitative concepts and manipulate numerical symbols. Using math problems is the best approach for this measurement.
>
> **In summary, reorganizing MMMU's questions to measure MLLMs' GIA is not feasible because its questions focus on domain-specific knowledge and do not meet the requirements of intelligence test questions.**
>
> [1] Yu W, et al. Mm-vet: Evaluating large multimodal models for integrated capabilities, 2023
>
> [2] MME: A Comprehensive Evaluation Benchmark for Multimodal Large Language Models, 2023
>
> [3] Yue X, et al. Mmmu: A massive multi-discipline multimodal understanding and reasoning benchmark for expert agi. CVPR 2024
>
> [4] Lu P, et al. Mathvista: Evaluating mathematical reasoning of foundation models in visual contexts, 2023
>
> [5] CoMix: A Comprehensive Benchmark for Multi-Task Comic Understanding, 2024.

---

> > ### Comment · Reviewer_1Zrf · 2024-11-22
> >
> > Thank you for your reply.
> >
> > First, it appears that II-Bench [1] contains questions related to Comic Problems.
> >
> > Second, you mentioned: *In contrast, M3GIA, except for the Gq cluster (Math), consists of cognitive test questions designed to minimize reliance on prior knowledge and focus solely on intelligence.*
> > Could you kindly list out which question types in M3GIA are not related to prior knowledge?
> > In my understanding, they are "Visualization, Concept Formation, Picture Recognition, Syllogism Problems, and Raven's Matrices." In total, there are 5 question types.
> > Considering there are 16 question types in M3GIA as showed in figure 3, it seems that the majority of question types are related to prior knowledge, such as Math, Text Understanding, and world knowledge, among others.
> >
> > Third, regarding the IQ test versus Task-Oriented test, I find your point very insightful. However, I would appreciate it if you could elaborate on why/when we need an IQ test for LMMs. For instance, if we aim to select the best LMM to deploy for general domain chatting or a specific downstream task, why would we choose a model with a high IQ? Does the IQ score of a model strongly correlate with its performance on downstream tasks? Or does the output from a High IQ model tend to be more preferred by human users?
> > I personally believe it would be beneficial if you could demonstrate that, compared to the MMMU score, the IQ score has a better correlation with the Chatbot Arena [2] rank (considering that the Chatbot Arena rank to some extent reflects human user preference).
> >
> > [1] II-Bench: An Image Implication Understanding Benchmark for Multimodal Large Language Models https://arxiv.org/pdf/2406.05862
> > [2] Chatbot Arena https://lmarena.ai/

---

> ### Author Response · Authors · 2024-11-24
> **Question 1**
>
> Thank you for your reply. We appreciate that you find our point regarding the IQ test versus Task-Oriented test very insightful.
>
> > **Which question types in M3GIA are not related to prior knowledge? It seems that the majority of question types are related to prior knowledge**
>
> Sorry, the term "prior knowledge" we used in that sentence is inaccurate. What we originally intended to express was the term used in the previous text: **professional domain-specific knowledge**, rather than general everyday prior knowledge (as the context of this reply is primarily targeted at MMMU).
>
> - **If we discuss domain-specific knowledge**, then only the Math section of M3GIA would involve it, with specific question types including:
>
>   1. Math facts
>   2. Algebra & Geometry
>   3. Applied math problems
>
>   Number of questions: 60 * 6 = 360 (20% of M3GIA)
>
>   As mentioned earlier, these question types were specifically chosen because they were designed to measure the Gq factor, and the definition of the Gq factor itself is closely related to domain-specific mathematical knowledge.
>
> - **If we expand the scope to ordinary everyday common knowledge**, then indeed, only the following types of questions are completely unrelated to common knowledge (as you mentioned):
>
>   1. Visualization
>   2. Picture Recognition
>   3. Concept Formation
>   4. Syllogism Problems
>   5. Raven's Matrices
>   6. Number Series
>
>   Number of questions: 115 * 6 = 690 (38.3% of M3GIA)
>
> - **The remaining question types include:**
>
>   1. General Information
>   2. Oral Vocabulary
>   3. Logo Problem
>   4. Real-world Spatial
>   5. Readings
>   6. Comic Problem
>   7. Real-world Reasoning
>
>   Number of questions: 125 * 6 = 750 (41.7% of M3GIA)
>
>   These questions need to be discussed in two parts:
>
>   - General Information, Oral Vocabulary, Logo Problem
>   - Others
>
>   For the latter, the model does implicitly rely on everyday common knowledge when solving problems. For instance, when analyzing a comic strip, the model would first need to understand very basic concepts like "boy", "girl" or "sad".
>
>   **However, this level of prior knowledge does not affect the validity of the questions as proper cognitive test items.** In fact, this approach is very common in professional psychological testing [1][2]. In the field of psychometrics, tasks are typically composed of control factors (i.e., baseline abilities) and target factors [3]. The goal is to control for baseline ability requirements while focusing on measuring target abilities, ignoring the influence of baseline abilities.
>
>   Because these problems do not rely heavily on specialized domain-specific prior knowledge like those in MMMU, **the basic knowledge involved is not a dominant factor in determining whether the problem can be solved.**
>
>   For the former, they were specifically designed to assess world knowledge under the Gc factor, so naturally, prior knowledge would be needed to solve them. In fact, General Information and Oral Vocabulary are also question types used in WJ-IV cognitive testing to assess Gc. The reasoning for including the Logo Problem as a Gc-related question type has been elaborated in the appendix.
>
> A particularly unique question type is Number Series, which you did not mention. Number Series is widely used in WJ-IV cognitive testing to assess RQ and I. It is often assumed to rely on some mathematical knowledge. However, in WJ-IV, following the "primary factor" principle mentioned earlier, it is not categorized under Gq. This is because Number Series requires very little mathematical knowledge; its main challenge lies in applying inductive reasoning to identify patterns, without involving formulas or theorems emphasized by Gq. We followed WJ-IV’s approach in this regard.
>
> [1] Wechsler D, Kodama H. Wechsler intelligence scale for children
>
> [2] Roid G H, Barram R A. Essentials of Stanford-Binet intelligence scales (SB5) assessment
>
> [3] Schrank F A, Decker S L, Garruto J M. Essentials of WJ IV cognitive abilities assessment

---

> ### Author Response · Authors · 2024-11-24
> **Question 2**
>
> > **Does the output from a High IQ model tend to be more preferred by human users?**
>
> Thank you very much for your insightful suggestion -- **it led us to fascinating conclusions!**
>
> We performed linear regression to calculate the R-squared correlation between various models' scores on **Chatbot Arena** and their **GIA scores** from M3GIA. We also compared these correlations with scores obtained from traditional task-oriented benchmarks, such as MMMU, MMBench, MM-Vet, and OCR-Bench. The results are as follow:
>
> Models|GPT-4o|Gemini-1.5-Pro|Gemini-Pro|Claude-3-Sonnet|Claude-3-Haiku
> |:----|:----:|:----:|:----:|:----:|:----:|
> Arena Score|1361|1301|1111|1201|1079
> M3GIA*|92.4|78.1|69.9|72.5|71.2
> MMBench|80.5|73.9|69.7|81.7|57.1
> MMMU|69.2|60.6|49.0|66.4|49.7
> OCRBench|80.5|75.4|68.0|64.6|65.8
> MM-Vet|75.1|64.0|58.6|51.7|46.4
> Average Performance on 8 benchmarks**|71.5|64.4|54.1|53.5|51.5
>
> Benchmarks|M3GIA|MMBench|MMMU|OCRBench|MM-Vet|Average Performance
> |:--------|:----:|:----:|:----:|:----:|:----:|:----:|
> R-squared|**0.83**|0.25|0.61|0.75|0.57|**0.84**
>
> \* *The GIA scores are normalized results after setting the average human GIA scores for each language to 100.0. (as discussed in Sec4.2)*
>
> \** *We calculated the average scores **across 8 prominent benchmarks**, including MMMU, MMBench, MM-Vet, OCR-Bench, HallusionBench, AI2D, MMStar, MathVista*
>
> ## Key Findings:
>
> 1. **Strongest Correlation with Human Preference:**
> Our GIA score **indeed** demonstrated **the strongest correlation** with human preference scores on Chatbot Arena among all the benchmarks evaluated.
>
> 2. Benchmark Averaging as a Comparison:
>
> - Challenges with Benchmark Aggregation:
> In the current MLLM community, it is widely recognized that a single benchmark often fails to truly reflect model capabilities, leading to **significant gaps between benchmark scores and actual human experiences.** To address this, researchers commonly resort to averaging scores across multiple benchmarks, but this process is time-intensive and resource-heavy.
>
> - To validate the significance of M3GIA, we calculated the average scores of the models **across 8 prominent benchmarks**, including MMMU, MMBench, MM-Vet, OCR-Bench, HallusionBench, AI2D, MMStar, MathVista and found:
>
>   - The average score across these benchmarks exhibited a higher correlation with human preference scores compared to individual benchmark scores.
>
>   - Comparable Results: Interestingly, the correlation between the average benchmark score and human preference (R^2 = 0.84) is almost identical to the correlation between the M3GIA GIA score and human preference (R^2 = 0.83).
>
> - Conclusion:
>
>   **M3GIA achieves a level of correlation with human preferences equivalent to the aggregation of multiple benchmarks. Crucially, it achieves this with just a single, unified test suite, significantly simplifying the evaluation process and addressing the pain point of benchmarking complexity in the MLLM community.**
>
> P.S. -- The models we selected for this analysis are all the multimodal models currently featured on Chatbot Arena (the rest are purely language models).
>
> To further illustrate the intriguing conclusions, we visualized the results and included them in Appendix H.
>
> Thanks again for your insightful suggestion!

---

> > ### Comment · Reviewer_1Zrf · 2024-11-25
> >
> > I appreciate that the authors took my suggestion into consideration.
> >
> > However, it seems the arena score you used is from the general domain arena, which is not multimodal. You might consider using the arena score (vision) for conducting the correlation analysis, since M3GIA is a multimodal benchmark rather than a pure text benchmark.
> >
> > By the way, does the GIA score represent the average accuracy (ACC) shown in the last column of Table 1?

---

> ### Author Response · Authors · 2024-11-25
>
> > **Consider using the arena score (vision) for conducting the correlation analysis.**
>
> Thank you for your suggestions. We have added experiments using **Arena Score (Vision)** for correlation analysis and include more models this time. The results are as follows:
>
> Models|GPT-4o|Gemini-1.5-Pro|Claude-3-Sonnet|Claude-3-Haiku|GPT-4o-mini|Qwen2-VL-7B|MiniCPM-v 2_6
> |:----|:----:|:----:|:----:|:----:|:----:|:----:|:----:|
> Arena Score|1227|1220|1048|1000|1122|1053|975
> M3GIA (GIA score)*|92.4|78.1|72.5|71.2|75.6|74.3|65.6
> MMBench (Acc.)|80.5|73.9|81.7|57.1|75.9|83.0|81.8
> MMMU (Acc.)|69.2|60.6|66.4|49.7|60.0|54.1|49.8
> OCRBench (Acc.)|80.5|75.4|64.6|65.8|78.5|84.5|85.2
> MM-Vet (Acc.)|75.1|64.0|51.7|46.4|66.9|62.0|60.0
> Average Performance on 8 benchmarks**|71.5|64.4|53.5|51.5|64.1|63.3|60.5
>
> Benchmarks|M3GIA|MMBench|MMMU|OCRBench|MM-Vet|Average Performance
> |:--------|:----:|:----:|:----:|:----:|:----:|:----:|
> R-squared|**0.74**|0.03|0.53|0.02|0.55|**0.56**
>
> \** *We calculated the average scores across 8 prominent benchmarks, including MMMU, MMBench, MM-Vet, OCR-Bench, HallusionBench, AI2D, MMStar, MathVista*
>
> We observed that **M3GIA still exhibits the highest correlation with Arena Score.** Beyond this, we identified several intriguing findings:
>
> 1. OCRBench, which is task-specific and focus on a single capability, exhibit almost no correlation with the Arena Score, which reflects human preferences comprehensively. The result is reasonable.
>
> 2. Besides M3GIA, multi-task benchmarks tend to show better correlation with the Arena Score (mmvet, mmmu) compared to single-task benchmark. However, an exception is MMBench, where Qwen2-VL and MiniCPM-v2_6 emerge as significant outliers (please see the visualization in Appendix H2).
>
> We believe this partially reflects the possibility of whether models have hacked the benchmark. Considering that MMBench was proposed relatively early, and its data distribution is more accessible on the internet, such results align with expectations.
>
> This further highlights the importance of robust and diverse evaluation benchmarks like M3GIA in assessing model performance comprehensively and reliably. In summary, as the "IQ test" for MLLMs, M3GIA aligns more effectively with actual human experience.
>
> > **Does the GIA score represent the average accuracy (ACC) shown in the last column of Table 1.**
>
> **The GIA score in this study refers to the metrics used in the right part of Table 2.** Since Arena (Vision) only covers English and Chinese, the GIA score we used in this analysis is also limited to GIA_en and GIA_ch.
>
> **Essentially, GIA score can be understood as a refined measure obtained through Confirmatory Factor Analysis (CFA), which identifies the contributions of different factors to the overall GIA and assigns distinct weights to each factor.** This is in contrast to the traditional Acc metric, which assumes equal contributions of all dimensions to the total score. GIA score is a method rooted in the field of cognitive science [1][2]. The GIA score can largely reflect the Acc. metric while providing a more fine-grained assessment.
>
> As for why GPT-4o's GIA score in Table 2 differs from its score in this analysis, it is because this experiment uses the ChatGPT-4o-latest (2024-11-20) version.
>
> For details about the GIA score, please refer to Sec.4.1 (line 455-467), Sec.4.2 and Appendix.F "THE GIA METRICS"
>
> [1] Dubois J, Galdi P, Paul L K, et al. A distributed brain network predicts general intelligence from resting-state human neuroimaging data[J]. Philosophical Transactions of the Royal Society B: Biological Sciences, 2018, 373(1756): 20170284.
>
> [2] Kristanto D, Liu X, Sommer W, et al. What do neuroanatomical networks reveal about the ontology of human cognitive abilities?[J]. Iscience, 2022, 25(8).

---

> > ### Comment · Reviewer_1Zrf · 2024-11-26
> >
> > I have read the author’s response and appreciate their efforts during the rebuttal process.
> > I am currently leaning toward either accepting or rejecting the manuscript.
> > Since there are no 5.5 scores available, I will retain a score of 5.
> > For the area chair, you may interpret my score as 5.5 (neutral) rather than 5 (borderline reject).
> >
> > ---
> >
> > ### The main reasons for not giving a higher score:
> > 1. **Justification for the Need for an IQ Test for MLLM:**
> >    As I stated in a previous response (when/why we need a IQ test for MLLM), although the authors followed my suggestion and demonstrated some unique advantages of the IQ test (e.g., better alignment with human preferences compared to other task-oriented evaluations such as MMMU), these advantages are not well demonstrated in the main body of the current submission. The author may consider in the future version to largely expand the discussion on this point.
> >
> > 2. **Lack of Novelty in the Tasks:**
> >    As mentioned earlier, I personally believe that only five tasks in the IQ test are novel, while the remaining 11 types can be found in prior works like II-Bench, MMMU, MMBench, etc.
> >
> > ---
> >
> > ### The main reasons for not giving a lower score:
> > 1. **Introduction of CHC Theory:**
> >    The introduction of CHC theory is commendable, as it provides a solid taxonomy for building a systemic evaluation framework for MLLM.
> >
> > 2. **Contributions to Multilingual IQ Tests and Benchmarking:**
> >    The contributions of including a multilingual version of the IQ test and providing a new benchmark for MLLM are significant.

---

> ### Author Response · Authors · 2024-11-26
>
> Dear Reviewer 1Zrf,
>
> Thank you for your comments and for taking the time to engage with our submission.
>
> 1. Firstly, **we would like to kindly remind you that ICLR 2025 allows modifications to the main text during the rebuttal phase.** Regarding the justification for the need for an IQ test for MLLMs, **we are actually integrating this discussion into the main text with more formal analyses.** We aim to finalize and resubmit the updated manuscript before November 27th AoE. If this inclusion addresses your concerns and strengthens the manuscript, we hope you might consider revisiting your scoring.
>
> 2. Additionally, given the extended rebuttal deadline to December 3rd, we would like to take this opportunity to clarify the concerns you raised that we respectfully disagree with:
>
>     > **I personally believe that only five tasks in the IQ test are novel, while the remaining 11 types can be found in prior works like II-Bench, MMMU, MMBench, etc.**
>
>     **With respect, we strongly believe that this should not be a point to reject the paper.**
>
>     When examining current renowned MLLM benchmarks, such as *MMBench (ECCV 2024)*, *MMVet (ICML 2024)*, and *SeedBench-IMG (CVPR 2024)*, one can observe *it is very common that many task types within these benchmarks overlap with those from prior works.* (Please refer to the "*" at end of this reply for specific examples.)
>
>     Despite this, their novelty is not undermined -- they have distinct task organizational structures and purpose -- *what distinguishes a benchmark is not merely its tasks but the framework, methodology, and purpose it serves.* It is reasonable -- and often unavoidable -- that similar task types appear across different benchmarks. Borrowing similar types of tasks only serves as part of their own unique goal. If such task types indeed align well with their own frameworks, and the specific data is brand new to the community, why not use them?
>
>     For M3GIA, its greatest novelty lies in its organization under the CHC theoretical framework, which focuses on evaluating the cognitive abilities of MLLMs. *The selection of our task types, including those that may overlap with other benchmarks, was deliberate, as these tasks **indeed align very well with the CHC structure and are particularly suited to measuring intelligence.*** (as discussed)
>
>     **We believe excluding question types merely for the sake of novelty, rather than their suitability for the overarching goal, would be counterproductive -- The best question types are those that best serve the unique goals and organizational structure of the benchmark, rather than deliberately pursuing superficial innovation in its form.**
>
>     Furthermore, upon deeper examination, you will notice that the degree of overlap between tasks in other benchmarks (as mentioned above) is far greater than the overlap between M3GIA and these benchmarks.
>
> If there are further specific concerns or aspects of M3GIA that you believe need improvement, we are more than willing to address them.
>
> \* **Examples:**
> - Perception-Count in MME and Object Localization in MMBench (For example, the model is asked: "How many apples are there in the image? And how many bananas are there?")
> - OCR in MMVet, OCR in MMBench, OCR in MME and OCRBench.
> - Math in MMMU and math in Mathvista
> - Structuralized Image-text Understanding in MMBench and ChartQA, TextQA
> - Handwriting math problems in MMVet and other benchmarks that include math.
> - Scene Understanding in SEED and Perception-Scene in MME
> - Instance Identity in SEED and Identity Reasoning in MMBench and so on.
>
> ...
> Too numerous to list exhaustively.

---

> ### Author Response · Authors · 2024-12-01
> **Gentle Reminder: Review of Rebuttal & Final Score**
>
> Dear Reviewer 1Zrf,
>
> We sincerely appreciate your valuable comments. We have carefully considered your feedback and **submitted the revised version of our paper**, including the justification for the need for an IQ Test for MLLM.
>
> As we approach the conclusion of the discussion phase, could we kindly know if the responses have addressed your concerns and if further explanations or clarifications are needed? If this inclusion addresses your concerns and strengthens the manuscript, we hope you might consider revisiting your scoring.
>
> Your time and efforts in evaluating our work are appreciated greatly. If you have any concerns, we are very eager to engage in discussion with you to address any potential misunderstandings.
>
> Best,
>
> Paper 7079 Authors

---

### Official Review · Reviewer_o6ax · 2024-11-03

**Soundness:** 3
**Presentation:** 3
**Contribution:** 2
**Rating:** 5
**Confidence:** 5

**Summary:**

This paper introduced a cognitive-driven multi-lingual and multi-modal benchmark, dubbed M3GIA, to evaluate the general intelligence ability of multi-modality large language models (MLLMs). Based on the Cattell-Horn-Carroll (CHC) model from cognitive science, the authors build a benchmark including 1.8K QAs annotated by native speakers in five languages. Experiments and analysis on 24 MLLMs show the significant disparities between MLLMs and human performance.

**Strengths:**

- Based on the CHC theory, this paper brings a new perspective to the MM community for constructing multi-modal benchmarks aimed at evaluating modern MLLMs in terms of human-level intelligence. The background and taxonomy of the CHC theory are clear and meaningful.
  - As a benchmark, the multi-modal QAs annotated by humans are of high quality and useful.
  - The evaluation of both open-source and closed-source MLLMs is extensive and thorough.

**Weaknesses:**

- Though starting from a new perspective of the CHC theory, this paper still evaluates the widely adopted capabilities of MLLMs that have been investigated in previous benchmarks, such as Visual-Spatial Processing, Knowledge, Math Facts, and Text Reading. For example, the MM-vet benchmark builds QAs related to the capabilities of OCR, Math, Knowledge, and Language Generation, using LLMs as examiners to evaluate open-ended generations. The performance of MLLMs in Table 1 also demonstrates a consistent trend between M3GIA and other general multimodal benchmarks, rather than revealing distinct findings.
  - This paper spend extensive content to introducing the CHC model within the main content. However, one point still remains unclear to me: how does the CHC model affect the capabilities of MLLMs? In other words, what specific attributes or behaviors would a powerful MLLM, grounded in CHC theory, exhibit? Are there any case studies or pilot experiments that illustrate the significance of this influence?
  - The paper is missing detailed statistical information about the proposed benchmark, such as the number of images per category and the average number of words in the generated questions.
  - The paper’s experimental section appears to be incomplete due to the absence of results for the few-shot setting.

**Questions:**

- For the Human Performance Baseline, I believe that these results are important for reflecting the difficulty of the created benchmark. What are the educational levels of the participants, and how is the quality of the created questions ensured?

---

> ### Author Response · Authors · 2024-11-22
> **Reply window 1**
>
> Apologies for the delayed response. We are conducting experiments to address your concern. We're glad to hear that you recognize M3GIA presents an interesting perspective on benchmark construction. We address your concerns below.
>
> > **Concern1: this paper still evaluates the widely adopted capabilities of MLLMs that have been investigated in previous benchmarks**
>
> With respect, we disagree.
> Our cognitive-based question types, such as Concept Formation, Visualization, Picture Recognition, and Syllogism Problems, have not been systematically featured or examined in previous benchmarks.
> In other words, **no previous benchmark has systematically included and assessed these cognitive-based question types within a single set of test questions.**
>
> Although some previous benchmarks feature question type names that may appear similar to ours -- such as "Perception" in MME and "Spatial Awareness" in MM-Vet -- the similarity lies only in the names, as **the actual question types they assess are quite different from ours.**
>
> For example, their visual assessments often focus on traditional CV tasks, such as celebrity recognition or scene recognition. In contrast, our Visual-Spatial cluster uses IQ test questions designed for human cognitive testing. For instance, in our Visualization-Block Rotation test, the model is asked to identify the rotated 3D block that matches the original 3D block -- entirely new question types that have not appeared in any prior benchmarks. Compared to theirs, our question design rigorously references human intelligence test questions.
>
> Additionally, taking MM-Vet as an example, while its question types include OCR, Math, Knowledge, and Language Generation -- with its Math and Knowledge related to our Gq and Gc clusters -- **it does not comprehensively cover all the cognitive dimensions we examine**, such as Gv, Gf, and Grw. Not only MM-Vet, but to the best of our knowledge, no existing benchmark has approached this from an Intelligence Theory perspective and fully covered all the factors we assess within a single set of test questions.
>
> You may also refer to General Response to see “How does M3GIA essentially differ from other MLLM benchmarks”.
>
> > **Concern2: what specific attributes or behaviors would a powerful MLLM, grounded in CHC theory, exhibit.**
>
> Sorry for the confusion, in fact, the answer to this question lies in the definitions of the CHC factors themselves.
>
> Specifically:
> - **Gc** is the breadth and depth of acquired knowledge of culture that is incorporated during general life experiences. *So, a model with strong Gc should possess a broad common knowledge base and the ability to apply and recall this knowledge*[1]
>
> - **Gf** is the broad ability involved in reasoning, forming concepts, and solving problems. *So, a model with strong Gf should have:*
>   - strong ability to observe underlying pattens or rules (I).
>   - strong capacity to reason logically using known premises step by step (RG).
>   - strong ability to reason with numbers, mathematical relations, and operators (RQ).
>
> - **Gv** is the ability to perceive visual stimuli and perform spatial imagination.[2] *So, a model with strong Gv should have a strong ability to encode visual input, and achieve tasks that require spatial imagination.*
>
> - **Grw** is the depth and breadth of knowledge and skills related to written language. *So model with high Grw should read with little effort and have a strong ability in understanding the potential relationship between texts* [1].
>
> - **Gq** is the depth and breadth of knowledge about mathematics such as symbols, operations, computational procedures. *So, a model with strong Gq should have a strong ability to comprehend quantitative concepts and to manipulate numerical symbols.*
>
> For detailed definitions of the CHC factors, please refer to Tab.2,3 in the Appendix. The case study in Appendix.I also demonstrates the differences between models in these abilities.
>
> **In addition, we conducted a series of ablation study and discovered a meaningful conclusion:**
> - We use Qwen1.5-7B as the base model and trained a series of MLLMs using different ViTs as vision encoders, all with the same data.
> - **We found that the vision encoder does indeed impact Gv performance, and the effect of the sample size seen by the vision encoder outweighs the impact of its parameter size on Gv.**
>
> |Vision Encoder|ViT-L/14|ViT-L/14|ViT-H/14|ViT-G/14
> |:----:|:----:|:----:|:----:|:----:|
> Params.|303M|303M|632M|3000M
> Samples seen|13B|32B|32B|34B
> Gv Acc.|0.308|0.383|0.375|0.392
>
> This can be explained as: expanding the visual vocabulary enables the LLM to encode visual inputs with greater granularity, which helps solve complex abstract visual reasoning tasks
>
> This suggests that when designing the self-encoder module for MLLMs, data may be more critical than merely scaling up the model size. We will add the experiment in the paper.
>
> [1] Essentials of WJ IV Cognitive Abilities Assessment
>
> [2] The Woodcock--Johnson IV

---

> ### Author Response · Authors · 2024-11-22
> **Reply window 2**
>
> > **Concern 3: The paper is missing detailed statistical information about the proposed benchmark, such as the number of images per category and the average number of words in the generated questions.**
>
> Thanks for the feedback, we will place the specific data statistics in Appendix E "DATA CURATION PROCESS".
>
> |QuestionTypes|Average number of words|Number of images
> |:----------|:-----:|:------:|
> |General Information|12.3|120
> |Oral Vocabulary|17.5|90
> |Logo Problem|16.6|90
> |Visualization|65.9|90
> |Picture Recognition|24.0|180
> |Real-world Spatial|10.3|90
> |Readings-VL|257.3|60
> |Readings-text|303.6|0
> |Comic Problem|14.7|90
> |Math Facts|13.2|60
> |Algebra|16.0|90
> |Geometry|25.9|60
> |Applied Problem|28.5|60
> |Number Seres|28.6|120
> |Concept Formation|11.0|120
> |Raven's Matrices|56.0|60
> |Syllogism Problem|84.5|120
> |Real-world Reasoning|113.3|120
> |Total|53.9|1,620
>
> > **Concern 4: The paper’s experimental section appears to be incomplete due to the absence of results for the few-shot setting.**
>
> Thanks for your feedback. We will add the few-shot evaluation result and analysis in the supplementary material, Appendix H2 'Few-shot Evaluation.' The prompts we used are placed at the end of the appendix. The experimental results are as follows.
>
> |model|shots|overall|Gf (I)|Gf (RG)|Gf (RQ)|Gf (overall)|Gc|Gq|Grw|Gv
> |:---:|:---:|:---:|:----------:|:----------:|:----------:|:------------:|:----:|:---:|:---:|:---:|
> |llava1.6 7b|0|27.33|18|27.5|9.09|17.6|38.75|8.33|42.5|23.33
> ||1|18.33|10|32.5|3.63|16|25|8.33|32.5|10
> llava1.6 13b|0|31.33|18|32.5|9.09|19.2|51.25|8.33|42.5|26.66
> ||1|22.66|12|42.5|1.81|19.2|36.25|5|32.5|14.16
> qwen-vl-base|0|36.33|28|32.5|30.91|30.4|50|26.66|40|36.66
> ||1|29.66|16|20|16.36|17.6|41.25|20|35|34.16
> ||5|27.66|14|15|16.36|15.2|45|18.33|20|31.66
> qwen-vl-chat|0|36.33|18|22.5|23.64|20|46.25|30|52.5|35.83
> ||1|35.33|24|17.5|18.18|18.4|45|21.66|45|38.33
> ||5|35.66|24|25|21.82|22.4|48.75|20|47.5|34.16
> qwen2-vl-7b|0|60.33|40|57.5|38.18|47.2|76.25|51.66|75|57.4
> ||1|61.33|50|60|40|52|76.25|53.33|75|53.33
> ||5|62.66|56|55|43.63|52.8|73.75|55|75|54.16
> gpt4o|0|67|58|85|50.90|64.8|90|58.33|62.5|53.33
> ||1|68|60|80|43.63|62.4|86.25|53.33|75|60
> ||5|68.66|62|72.5|45.45|60.8|86.25|58.33|80|58.33
>
> In our few-shot prompts, the images used are ones that do not appear in the test set, and we ensure that the few-shot examples maintain the same distribution as the corresponding question types.
>
> In the experiment, we observed a somewhat counterintuitive phenomenon: the few-shot approach did not bring a significant improvement in the model's performance on M3GIA. In fact, it even had a counterproductive effect on some weaker early models (e.g., LLaVA 1.6, QwenVL-Chat).
>
> We analyze that the reason for this is that we did not use few-shot prompts with Chain-of-Thought (CoT), but instead guided the models with question-answer pairs. This primarily strengthens the model's instruction-following ability, rather than its reasoning ability (especially for a challenging benchmark like M3GIA, simply providing questions and answers does not significantly help the model understand the problem-solving process).
>
> For stronger models, such as Qwen2-VL and GPT-4o, their instruction-following ability is already quite strong, and the few-shot approach brings limited gains. For weaker models (e.g., LLaVA 1.6, QwenVL-Chat), the interference brought by few-shot is even greater than the gains, which is why earlier benchmarks rarely used few-shot for measurement [1].
>
> We will further refine the analysis of this experiment in the Appendix H2 and provide the impact of CoT on the results before the final version.
>
> [1] Liu Y, Duan H, Zhang Y, et al. Mmbench: Is your multi-modal model an all-around player? ECCV 2025

---

> ### Author Response · Authors · 2024-11-24
> **Gentle Reminder: Review of Rebuttal & Final Score**
>
> Dear Reviewer o6ax,
>
> We sincerely appreciate your valuable comments. We have carefully considered your feedback and made corresponding improvements to our paper.
>
> As we approach the conclusion of the discussion phase, could we kindly know if the responses have addressed your concerns and if further explanations or clarifications are needed? Your time and efforts in evaluating our work are appreciated greatly. If you have any concerns, we are very eager to engage in discussion with you to address any potential misunderstandings.
>
> Best,
>
> Paper 7079 Authors

---

### Official Review · Reviewer_5FWL · 2024-11-03

**Soundness:** 2
**Presentation:** 2
**Contribution:** 2
**Rating:** 3
**Confidence:** 3

**Summary:**

This paper presents a benchmark M3GIA which claims to act as the first “IQ test” for multimodal large language models (MLLM). It is built based on five cognitive factors from the Cattell-Horn-Carroll Model of Intelligence. It includes VQA/text-format questions from tasks like oral vocabulary, concept formation, visualization, math, reading and etc. Besides English, it also includes other languages such as Chinese, French, Spanish, Portuguese and Korean. The authors evaluate their benchmark on a number of API-based and open-source models across different scales as well as human participants. They observe that the best MLLM (GPT4-o) can reach the lower boundary of human performance in English.

**Strengths:**

The paper presents an interesting perspective for constructing benchmarks and suggests we can design benchmarks based on previous cognitive science studies. The contribution of the new resources can be helpful and raise more questions and considerations about benchmark design. They also provide an initial performance analysis of some of the existing models, which can be used as a reference for future research.

**Weaknesses:**

While the authors claim that M3GIA can serve as an IQ test for MLLMs and have built this benchmark based on existing cognition theory, I find it hard to conclude generally that “most advanced MLLM reaches the lower boundary of human intelligence in English”. There are many different categories of questions collected in this benchmark and they can fall under different cognitive factors. However, it is unclear what control factors are in place during the data collection and evaluation process: why this specific type of question is chosen? How are the variances of questions controlled across languages? How broad/narrow is the topic tested in each domain? What are the sample demographics of the annotators? Given there are only 300 questions tested per language, it’s hard to prove that the human responses collected represent the lower bound of human intelligence.

**Questions:**

- Can you provide more details about how you decide the question category under each factor and how is each question selected for each category? Is there any data filtering or quality inspection process from experts to determine whether each question is easy/hard enough to be included?
- How does this dataset differ from other reasoning benchmarks besides it is “cognition-inspired”? If the importance lies in its originality, why do you also include datapoints from other datasets?

---

> ### Author Response · Authors · 2024-11-19
> **Reply window 1**
>
> Thank you for your time and effort in reviewing our paper. We're glad to hear that you recognize M3GIA presents an interesting perspective on benchmark construction. We address your concerns below.
>
> > **Concern: Why are these specific types of questions chosen?**
>
> Thanks for the question. In fact, this was the aspect we dedicated the most effort to when designing M3GIA.
>
> - We choose these types of question since their contents directly point to the definitions of the CHC factors.
> - To ensure that M3GIA maintains professionalism as a cognitive science test, we adhered to the question designs of the well-recognized WJ-IV[1] for each CHC factor.
>
> In WJ-IV, **each question type is specifically crafted according to the definition of a specific sub-factor within a CHC factor.**
>
> The content of the questions aligns so closely with the corresponding CHC factor definitions that their selection to assess these factors feels intuitive.
>
> We do understand that readers may raise this question because they may not be familiar with the WJ-IV and definitions of CHC. So, we dedicated a new section in Appendix.D to further elaborate on this.
>
> **Please check out the updated version of Appendix and refer to Sec. D, "Connection between the Question Types and the CHC Factors" for details (especially Table2, 3).** We also attach the table in the next reply window.
>
> > **Concern: How are the variances of questions controlled across languages?**
>
> We established a strict pipeline to ensure the consistency across language. We apologize for overlooking a detailed explanation of this in the main text, and will add the content to the paper.
> - Difficulty consistency:
>   - After each annotator created questions, the questions were tested by 3 additional annotators in that language, who were not provided with the answers and were asked to rate each question’s difficulty into 1 of 5 difficulty levels: A (very easy) to E (very difficult). We filtered out questions that two or more reviewers consistently rated as too easy (A) or too hard (E) and maintained consistency in the number of B, C, D-level questions across languages.
>   - Following this initial screening, the questions were reviewed by the psychology expert of our team. The expert further excluded questions deemed too easy or difficult and adjusted the proportions of B, C, D-level questions as necessary. As a result, the distribution of difficulty levels across languages is nearly identical. Please see Appendix.E2 for detailed statistics.
>   - According to our final human evaluation, the average scores across the six languages were nearly identical, which empirically validates the effectiveness of our approach.
> Laguage|Ch|Sp|Pt|Ko|Fr|En
> |:----:|:----:|:----:|:----:|:----:|:----:|:----:|
> Acc.|0.790|0.757|0.737|0.777|0.753|0.740
>
> - We also ensure the topics covered under the same question type across languages were as aligned as possible. For instance, in math application problems, if an English question involved ticket purchasing, we would also include a ticket-purchasing question for each language, tailored to the context of the respective country.
>
> You may find more details about data filtering and quality control in Appendix A.2, E.1 and E.2.
>
> > **Concern: Given there are only 300 questions tested per language, it’s hard to prove that the human responses represent the lower bound of human intelligence.**
>
> With respect, we disagree. As fully discussed in the main paper (line 315-323), 300 is a sufficient and appropriate number for testing human intelligence, which already takes 5-6 hours to complete.
> > Research by Converse & Presser (1986) indicates that prolonged tasks can degrade response quality... we determine the number of questions based on findings from Burisch (1997), which revealed that in cognitive assessments, extending a scale beyond a certain limit can actually undermine its validity. Interestingly, the validity plateaus when the number of items in a subtest hits 15. Considering our 18 subtests, we settled on incorporating 300 questions per language (>15x18 = 270) to guarantee a thorough evaluation.
>
> Following this psychological protocol, we found that, even the best-performing model GPT-4o, failed to surpass human in cognitive measurement, but its GIA score barely fell within the error bar range of human performance.
> ||Highest (human)|Lowest (human)|Avg.(human)|Std.(human)|Error bar|Lower boundary|GPT-4o
> |:----:|:----:|:----:|:----:|:----:|:----:|:----:|:----:|
> GIA Score|20.740|10.172|16.014|2.248|[13.766, 18.262]|13.766|13.847
>
> We indeed obtained this result and reported it honestly, ensuring the greatest possible validity in the test design (as above mentioned). To avoid controversy, we have revised the statement to "within the scope of our sample, the most advanced MLLM barely reaches the lower boundary of ..."
>
> > **Concern: How does M3GIA differ from other reasoning benchmarks besides it is “cognition-inspired”?**
>
> Please see reply window 3 or general response.

---

> ### Author Response · Authors · 2024-11-19
> **Reply window 2**
>
> > **Concern: Why are these specific types of questions chosen?**
>
> This is a supplement to the previous reply, where we mentioned:
> > The content of the question types aligns so closely with the corresponding CHC factor definitions that their selection to assess these factors feels intuitive.
>
> This table shows the close connection between the content of the question types and the definitions to the CHC factors. (It is also the Table2 in the updated Appendix.)
>
> Question Types|CHC~(sub-factor) Definition|Content of the question
> |:----|:----|:----|
> General Information|Gc~(K0): The store of language-based or verbal declarative (knowing what) and procedural (knowing how) knowledge acquired during general life experiences.|The model is presented with an image and is asked, “Where would you find [the object] in the picture?” or “What would you do with [the object]?”
> Oral Vocabulary|Gc~(VL): Knowledge of the definitions of words and the concepts underlie them.|The model is provided with a word and is asked to choose its synonym or antonym.
> Visualization|Gv~(Vz): The ability to perceive complex patterns and mentally simulate how they might look when transformed (e.g., rotated, changed in size, partially obscured, and so forth).|It consists of two subtests: In Block Rotation, the model is asked to identify the rotated 3D block that match the original 3D block. In Spatial Relations, the model is required to identify pieces that form a complete target shape.
> Picture Recognition|Gv~(MV): The ability to remember and identify complex images, also known as Visual Memory.|The model is presented with a shape, and is asked to identify the shape within a field of distracting shapes.
> Reading|Grw~(RC): The ability to understand written discourse.|The model is required to answer questions related to the main ideas of long articles (4-6 paragraphs) or the relationships between paragraphs.
> Math Facts|Gq~(KM): Range of general knowledge about mathematics. This factor is about “what” rather than “how” knowledge.|The questions focuses on the model’s acquired knowledge about symbol and geometry, covering from elementary to university level. It doesn't rely on using mathematical knowledge for complex reasoning, but rather focus on the knowledge itself.
> Algebra&Geometry|Gq~(A3): Measured (tested) mathematics achievement. The full name of A3 is Mathematical Achievement.|Unlike math facts problem which can be directly answered once the knowledge is acquired, these problems require a further reasoning process. We source the questions from authentic exam papers across the six countries to measure the Mathematical Achievement factor.
> Number Series|Gf~(RQ): The ability to reason, either with induction or deduction, with numbers, mathematical relations, and operators.|The model is presented a numbers series with one or more numbers missing. The model must determine the numerical pattern and provide the missing number.
> Concept Formation|Gf~(I): The ability to observe a phenomenon and discover the underlying principles or rules that determine its behavior.|It requires the model to examine a series of shapes or pictures and then formulate a rule, and then figure out the item that do not coincide with the rule.
> Raven's Matrices|Gf~(I): See above.|The model is asked to identify the missing element that completes a pattern. Patterns are presented in the form of a $4\times4$ or $3\times3$ matrix.
> Syllogism Problem|Gf~(RG): The ability to reason logically using known premises and principles. This ability is also known as deductive reasoning or sequential reasoning.|It is a classic form of deductive reasoning, where the model is asked to decide which of the given conclusions logically follows from the two given statements.

---

> ### Author Response · Authors · 2024-11-21
> **Reply window 3**
>
> > **Concern: How does this dataset differ from other reasoning benchmarks besides it is “cognition-inspired”?**
>
> As discussed in 081-092, other reasoning benchmarks like mmbench, mme, TextVQA, HallusionBench, mathvista, and SeedBench are task-oriented, focusing on one or several specific applied tasks. *Their objective is to evaluate model performance on practical application tasks themselves* (e.g., object attribute recognition, OCR, chart-solving, etc.).
>
> Their approach faces a significant issue that cannot be ignored (as discussed in line 084--086): **they struggle to provide a justified answer to 'why these specific ability dimensions were chosen for evaluation'** as their selection of ability dimensions is subjective and lacks a solid cognitive science underpinning.
>
> **In contrast, M3GIA does not primarily care about the model's performance in any specific application capability itself. Instead, our questions directly points to the definitions of the CHC factors**, reflecting the model’s performance across these dimensions of human cognition. The metrics it assesses directly correspond to relevant cognitive abilities, unlike other benchmarks where a given task might require multiple, interwoven cognitive abilities that are difficult to decouple.
>
> At the same time, these cognitive-based question types, such as Concept Formation, Visualization, Picture Recognition, Syllogism Problem have not been systematically featured or examined in previous benchmarks.
>
> In other words, **no previous benchmark has systematically included and assessed these cognitive-based question types within a single set of test questions.**

---

> ### Author Response · Authors · 2024-11-24
> **Gentle Reminder: Review of Rebuttal & Final Score**
>
> Dear Reviewer 5FWL,
>
> We sincerely appreciate your valuable comments. We have carefully considered your feedback and made corresponding improvements to our paper. These include a detailed discussion on why these specific types of questions were chosen, how the variances of questions are controlled across languages, and a clarification on how M3GIA distinguishes itself from other benchmarks.
>
> As we approach the conclusion of the discussion phase, could we kindly know if the responses have addressed your concerns and if further explanations or clarifications are needed? Your time and efforts in evaluating our work are appreciated greatly. If you have any concerns, we are very eager to engage in discussion with you to address any potential misunderstandings.
>
> Best,
>
> Paper 7079 Authors

---

> ### Comment · Reviewer_5FWL · 2024-11-26
>
> Can you remind me what the metric for measuring difficulty consistency is in the first table? Are these scores from humans or models?
>
> Regarding the comment "Given there are only 300 questions tested per language, it’s hard to prove that the human responses represent the lower bound of human intelligence.", I would like to thank the authors for clarification on human cognitive load when taking the test. The number of questions may suffice for human testers, however, models will not experience the cognitive load factor, and therefore same conclusion will not hold. I was wondering if the authors have considered this perspective?

---

> ### Author Response · Authors · 2024-11-29
> **Reply [1|2]**
>
> Thank you for your reply! We address your questions below.
>
> > **Question 1: What the metric for measuring difficulty consistency is in the first table? Are these scores from humans or models?**
>
> **It is the scores from human participants.** As we stated: *According to our final **human evaluation**, the average scores across the six languages were nearly identical, which empirically validates the effectiveness of our approach.*
>
> **Here, we should avoid using model scores for measuring difficulty consistency**, as models inherently exhibit varying capabilities across different languages, making their scores unsuitable for difficulty alignment.
>
> Instead, we rely on the scores of human participants, who were recruited with consistent age and educational backgrounds across all languages (As shown in the table below). This alignment of human participant scores is both meaningful and methodologically sound for ensuring cross-linguistic difficulty consistency.
>
> Age|En|Ch|Fr|Sp|Pt|Ko
> |:----:|:----:|:----:|:----:|:----:|:----:|:----:|
> (12, 18]|0.12|0.15|0.11|0.13|0.09|0.10
> (19, 25]|0.28|0.32|0.27|0.28|0.26|0.30
> (26, 35]|0.34|0.30|0.33|0.29|0.36|0.34
> (36, 55]|0.26|0.23|0.29|0.30|0.29|0.26
>
> Educational Background|En|Ch|Fr|Sp|Pt|Ko
> |:----:|:----:|:----:|:----:|:----:|:----:|:----:|
> $<$ Bachelor|0.31|0.29|0.36|0.35|0.38|0.30
> Bachelor       |0.44|0.46|0.42|0.44|0.41|0.44
> Master          |0.21|0.20|0.19|0.19|0.18|0.22
> Doctor          |0.04|0.05|0.03|0.02|0.03|0.04

---

> ### Author Response · Authors · 2024-11-29
> **Reply [2|2]**
>
> > **Question 2: Models will not experience the cognitive load factor, and therefore same conclusion will not hold. I was wondering if the authors have considered this perspective?**
>
> **Yes, we have *indeed* considered this issue early in our work and even invested substantial resources in attempting to address it by sampling 300 questions from a larger pool for human testers.**
>
> In fact, the early version of M3GIA contained 1,200 English questions. We sampled 25% of the questions from each category, balanced by difficulty levels (A-E), for human testing, while the full 1,200 questions were used to evaluate the models.
>
> We found that:
>
> 1. **The models' overall Acc. performance on the full 1,200-question set was almost identical to its performance on the 300-question subset.** This demonstrated that the smaller set of 300 questions was stable enough to achieve nearly the same measurement efficacy as the full 1,200 questions. (See the table below. *The numbers in the table represent the number of correctly answered questions for each cluster. For clarity and ease of comparison, the data for the 1,200-question set was normalized by dividing it by 4.*).
> Question Clusters|gpt_4o (en_1200)|gpt_4o (en_300)|gpt_4v (en_1200)|gpt_4v (en_300)|llava_v1.6_vicuna_34b (en_1200)|llava_v1.6_vicuna_34b (en_300)|Mini_Gemini_8B (en_1200)|Mini_Gemini_8B (en_300)
> |:------|:--------:|:--------:|:--------:|:--------:|:--------:|:--------:|:--------:|:--------:|
> Gc Cluster|42|46|40|44|40.5|44|39.25|43.00
> Gv Cluster|30.25|29|25.50|24|24.25|23|20|19.00
> Grw Cluster|28.75|28|28.5|27|25.25|25|23.25|24.00
> Gq Cluster|31|34|27.75|31|19.75|20|17.75|17.00
> Gf Cluster|62.5|61|58.5|58|30|28|37.5|36.00
>
> 2. **This approach introduced a critical issue: *inconsistencies between the test questions faced by models and humans.*** *This discrepancy created biases in the Confirmatory Factor Analysis (CFA) model used to compute the GIA score* (Section 3.3 and Appendix F: THE GIA METRICS). Results showed that the bias were beyond an acceptable range.
>
>    **When models are evaluated on 1,200 questions while humans are tested on the sampled set of 300 questions, the $R^2$ correlation between the GIA scores and the overall Acc. drops from 0.937 to 0.858.** (the $R^2$ metrics is obtained from the table below.)
>     > $R^2$ is a crucial metric for validating the reliability and interpretability of GIA scores in the field of psychometrics. Generally, $R^2$ is expected to fall within the range of (0.90–0.97), ensuring that the GIA score can largely reflect overall accuracy while providing a more fine-grained evaluation.
>
> Models|GIA Score (300-question set)|Acc. (300-question set)|GIA Score (1200-question set)|Acc. (1200-question set)
> |:------|:------:|:------:|:------:|:------:|
> gpt_4o|13.85|0.66|13.49|0.648
> gpt_4v|12.61|0.61|11.45|0.600
> llava_v1.6_vicuna_34b|11.47|0.47|10.76|0.465
> llava_v1.6_vicuna_13b|6.96|0.32|6.85|0.318
> llava_v1.6_vicuna_7b|6.75|0.29|6.72|0.284
> Mini_Gemini_34B|11.00|0.51|10.50|0.501
> Mini_Gemini_8times7B|11.05|0.50|12.04|0.492
> Mini_Gemini_13B|8.68|0.37|8.92|0.367
> Mini_Gemini_8B|9.32|0.46|9.70|0.459
> qwen72B-laion-clip-L|11.68|0.53|12.18|0.521
> qwen32B-laion-clip-L|10.58|0.48|11.48|0.469
> qwen14B-laion-clip-L|8.46|0.40|9.36|0.393
> qwen7B-laion-clip-L|8.56|0.41|9.96|0.406
> qwen1.8B-laion-clip-L|7.34|0.36|7.54|0.358
> qwen-vl|7.69|0.38|8.70|0.377
>
> After extensive discussions with the psychology experts on our team, we concluded that this approach is not feasible, as the discrepancies between test sets for models and humans compromise the robustness of the CFA model, making the approach insufficiently rigorous, and undermining the scientific integrity of the evaluation.
>
> Therefore, considering that 300 questions are robust enough to achieve results nearly identical to those obtained with 1,200 questions (Table 1), we opted to use the same 300 questions for both humans and models **to maintain fairness and ensure methodological rigor.**
>
> The current version was determined after thorough discussion and validation, considering multiple factors comprehensively.
>
> > PS: GIA score is a method rooted in the field of cognitive science [1][2]. Essentially, GIA score can be understood as a refined measure obtained through Confirmatory Factor Analysis (CFA), which identifies the contributions of different factors to the overall GIA and assigns distinct weights to each factor. This is in contrast to the traditional Acc metric, which assumes equal contributions of all dimensions to the total score. The GIA score can largely reflect the Acc. metric while providing a more fine-grained assessment.
>
> [1] Dubois J, et al. A distributed brain network predicts general intelligence from resting-state human neuroimaging data[J]. Philosophical Transactions of the Royal Society B: Biological Sciences, 2018, 373(1756): 20170284.
>
> [2] Kristanto D, et al. What do neuroanatomical networks reveal about the ontology of human cognitive abilities?[J]. Iscience, 2022, 25(8).

---

> > ### Author Response · Authors · 2024-12-01
> > **Gentle Reminder: Review of Rebuttal & Final Score**
> >
> > Dear Reviewer 5FWL,
> >
> > We sincerely appreciate your valuable comments. We have carefully addressed your proposed questions in details.
> >
> > As we approach the conclusion of the discussion phase, **could we kindly know if the responses have addressed your concerns?** If this inclusion addresses your concerns, we hope you might consider revisiting your scoring.
> >
> > Your time and efforts in evaluating our work are appreciated greatly. If you have any concerns, we are very eager to engage in discussion with you to address any potential misunderstandings.
> >
> > Best,
> >
> > Paper 7079 Authors

---

### Author Response · Authors · 2024-11-19
**General Response**

Dear Reviewers and AC,

Thank you all for your time and effort in reviewing our paper. We appreciate that reviewers:
1. Found our benchmark **presents an interesting perspective on benchmark construction** (5FWL) and brings a novel perspective for the MM community to design benchmarks aimed at evaluating modern MLLMs in terms of human-level intelligence (o6ax).
2. **Recognized the high quality and usefulness of our human-annotated multimodal QA data** (o6ax), and appreciated our practice of **using unpublished offline data to construct the benchmark to prevent data leakage** (1Zrf).
3. Found the background and taxonomy of the CHC theory clear and meaningful (o6ax), and **the use of the taxonomy is meaningful**, as it enables a more systematic evaluation (1Zrf).
4. Appreciated our practice of **including multiple language variants.** (1Zrf)
5. Recognized the evaluation of both open-source and closed-source MLLMs as extensive and thorough (o6ax).

We also thank 5FWL for recognizing the contribution of the new resources can be helpful and raise more considerations about benchmark design.

We are addressing each of your questions in the individual responses. Here we would like to emphasize our uniqueness compared to other benchmarks:

> **How does M3GIA essentially differ from other reasoning benchmarks?**

As discussed in 081-092, other reasoning benchmarks like mmbench, mme, TextVQA, HallusionBench, mathvista, and SeedBench are task-oriented, focusing on one or several specific applied tasks. *Their objective is to evaluate model performance on practical application tasks themselves* (e.g., object attribute recognition, OCR, chart-solving, etc.).

Their approach faces a significant issue that cannot be ignored: **they struggle to provide a justified answer to "why these specific ability dimensions were chosen for evaluation" as their selection of ability dimensions is subjective and lacks a solid cognitive science basis.**

**In contrast, M3GIA does not primarily care about the model's performance in any specific application capability itself. Instead, our questions directly points to the definitions of the CHC factors**, reflecting the model’s performance across these dimensions of human cognition. The metrics it assesses directly correspond to relevant cognitive abilities, unlike other benchmarks where a given task might require multiple, interwoven cognitive abilities that are difficult to decouple.

We would also like to emphasis that, our cognitive-based question types, such as Concept Formation, Visualization, Picture Recognition, Syllogism Problem have not been systematically featured or examined in previous benchmarks. In other words, **no previous benchmark has systematically included and assessed these cognitive-based question types within a single set of test questions.**

Thanks again for all the effort and time.

Best,

Authors

---

> ### Author Response · Authors · 2024-11-24
> **Update**
>
> Thanks to Reviewer 1Zrf for the insightful suggestion!
>
> We added an analysis of the correlation between our GIA score and human preference (Chatbot Arena[1]). The results demonstrate that, **compared to traditional task-oriented benchmarks, M3GIA exhibits the highest correlation with human ratings** (R-squared).
>
> Furthermore, its correlation is comparable to the average score across 8 prominent benchmarks, including MMMU, MMBench, MM-Vet, OCR-Bench, HallusionBench, AI2D, MMStar, MathVista, **highlighting M3GIA's potential to address the challenge in the MLLM community where a single benchmark often fails to align with true human experience, necessitating complex multi-benchmark evaluations.**
>
> This finding underscores the importance of conducting "IQ tests" for MLLMs and reveals their potential advantages over traditional benchmarks.
>
> *We have updated the supplementary materials and included the analysis and visualizations of this section in Appendix H2.*
>
> [1] Chiang W L, Zheng L, Sheng Y, et al. Chatbot arena: An open platform for evaluating llms by human preference[J]. arXiv preprint arXiv:2403.04132, 2024.

---

> ### Author Response · Authors · 2024-11-28
> **Revised version of the paper**
>
> Dear Reviewers and AC,
>
> **We have submitted the revised version of our paper** based on the reviewers' feedback. The new modifications have been highlighted in color for clarity, specifically:
>
> - Added the analysis of **the necessity of an IQ test for MLLMs**, M3GIA achieves a significantly stronger alignment with human preferences compared to traditional task-oriented benchmarks. (*Highlighted in blue*) | Reviewer 1Zrf
> - Included an **ablation study on different ViTs as vision encoders**, showing that the samples seen by the vision encoder have a greater impact on Gv performance than parameter size. (*Highlighted in orange*) | Reviewer o6ax
> - Provided **detailed explanations of the correspondence between our selected question types and CHC factors**, as well as measures to ensure cross-language consistency. (*Highlighted in purple*) | Reviewer 5FWL
>
> If you have any further questions or suggestions, we are more than happy to address them at any time.
>
> Best,
>
> Paper 7079 Authors

---

### Note · Authors · 2025-02-09

I have read and agree with the venue's withdrawal policy on behalf of myself and my co-authors.

---

### Meta-Review · Area_Chair_3GqJ · 2024-12-14

**Metareview:**

The paper proposes a broad benchmark to test models in different cognitively inspired tasks across modalities and skills.

Strengths:
Provides reasoning for validity.
Raises discussions

Weaknesses:
Small number of examples (perhaps a reliability test may account for it)
Possibly, overclaiming given the evidence
Missing data curation explanations
The relation to other benchmarks raised several questions (perhaps benchmark agreement testing and showing how the results and underlying traits captured differ and not only the motivation might help, [this](https://github.com/IBM/benchbench) tool and data might help)

The paper seems to provide an interesting contribution which I encourage the authors to revise and resubmit to a parallel venue.
Still, it got a lot of feedback that should merit an improved version.
Generally, ICLR calls for major changes during the rebuttal period considering those peer reviewed (despite it clearly not being the case). Here, they were not and as those are many. It is worth another round of peer review and revisions and evaluation to check the new state of the paper and experiments.

**Additional Comments On Reviewer Discussion:**

A massive rebuttal effort was put by the authors and ignored by reviewers despite calls (public and private) to engage.
Minor:
Note that the supplementary material is made for things like data, jsons, code etc. and appendix is allowed in ML venues after the references.

---

### Decision · Program_Chairs · 2025-01-22

Reject